# *Plasmodium berghei* liver stage parasites exploit host GABARAP proteins for TFEB activation
Jacqueline Schmuckli-Maurer[1], Annina F. Bindschedler[1,2], Rahel Wacker[1,2], Oliver M. Würgler[1], Ruth Rehmann[1], Timothy Lehmberg [3], Leon O. Murphy[3], Thanh N. Nguyen [4,5,6], Michael Lazarou[4,5,6], Jlenia Monfregola[7], Andrea Ballabio [3,7,8,9,10] & Volker T. Heussler [1] ✉

*Plasmodium*, the causative agent of malaria, infects hepatocytes prior to establishing a symptomatic blood stage infection. During this liver stage development, parasites reside in a parasitophorous vacuole (PV), whose membrane acts as the critical interface between the parasite and the host cell. It is well-established that host cell autophagy-related processes significantly impact the development of *Plasmodium* liver stages. Expression of genes related to autophagy and lysosomal biogenesis is orchestrated by transcription factor EB (TFEB). In this study, we explored the activation of host cell TFEB in *Plasmodium berghei*-infected cells during the liver stage of the parasite. Our results unveiled a critical role of proteins belonging to the Gamma-aminobutyric acid receptor-associated protein subfamily (GABARAP) of ATG8 proteins (GABARAP/L1/L2 and LC3A/B/C) in recruiting the TFEB-blocking FLCN-FNIP (Folliculin-Folliculin-interacting protein) complex to the PVM. Remarkably, the sequestration of FLCN-FNIP resulted in a robust activation of TFEB, reliant on conjugation of ATG8 proteins to single membranes (CASM) and GABARAP proteins. Our findings provide novel mechanistic insights into host cell signaling occurring at the PVM, shedding light on the complex interplay between *Plasmodium* parasites and the host cell during the liver stage of infection.

Malaria remains a pressing global health concern, with over 240 million estimated cases and more than 600,000 deaths occurring annually, primarily in tropical and subtropical regions (WHO 2022). *Plasmodium* parasites, the causative agents of malaria, are transmitted to humans through the bites of infectious female *Anopheles* mosquitoes[1,2]. Before establishing a symptomatic blood stage infection, *Plasmodium* parasites undergo an obligatory yet clinically silent liver stage (LS). Upon transmission to the vertebrate host, *Plasmodium* sporozoites migrate with the bloodstream passively to reach the liver. There, they breach the endothelial barrier and actively invade hepatocytes through invagination of the host cell plasma membrane, giving rise to a specialized compartment known as the parasitophorous vacuole (PV). The parasite resides within this PV throughout the LS development[3,4]. As the LS progresses, the parasite transforms into a trophozoite and then into a large liver schizont, generating thousands of daughter parasites, called

merozoites. These merozoites are eventually released into adjacent blood vessels within host cell plasma membrane-derived vesicles known as merosomes, which, upon reaching lung capillaries, rupture, liberating the merozoites to initiate the symptomatic blood stage infection by infecting erythrocytes[5].

The PV membrane (PVM) serves as the primary contact site between the parasite and the host cell cytoplasm. Despite its host cell origin, the PVM is substantially remodeled by the parasite through the export of parasite proteins, an essential process for successful parasite development[6]. Notably, the PVM also incorporates host cell proteins, including several associated with host cell autophagy, a cellular process that degrades and recycles cellular components[7,8]. Among these, the autophagy marker protein LC3B, which is a member of the ATG8 protein family, has been prominently observed on the PVM, marking the initiation of the *Plasmodium*-associated

[1]Institute of Cell Biology, University of Bern, Bern, Switzerland. [2]Graduate School for Cellular and Biomedical Sciences, University of Bern, Bern, Switzerland. [3]Casma Therapeutics, 400 Technology Sq, Cambridge, MA, 02139, USA. [4]Walter and Eliza Hall Institute of Medical Research, Parkville, Victoria, Australia. [5]Department of Biochemistry and Molecular Biology, Biomedicine Discovery Institute, Monash University, Melbourne, Australia. [6]Department of Medical Biology, University of Melbourne, Melbourne, Victoria, Australia. [7]Telethon Institute of Genetics and Medicine (TIGEM), Pozzuoli, Naples, Italy. [8]Department of Molecular and Human Genetics, Baylor College of Medicine, Houston, TX, USA. [9]Jan and Dan Duncan Neurological Research Institute, Texas Children's Hospital, Houston, TX, USA. [10]Medical Genetics Unit, Department of Medical and Translational Science, Federico II University, Naples, Italy. ✉e-mail: volker.heussler@unibe.ch

autophagy-related (PAAR) response, which encompasses various autophagic processes targeting *Plasmodium* LS parasites[9]. One such process is selective autophagy, where specific cellular components or organelles, such as damaged mitochondria or intracellular pathogens, are targeted for degradation. Intriguingly, while more than 90% of parasites early after infection are labeled with LC3B, only around 30-50% are eliminated[7], suggesting additional roles for LC3B.

Recently, we have shown that LC3B is incorporated into the PVM via a mechanism termed Conjugation of ATG8 proteins to Single Membranes (CASM)[10], which relies on the interaction between V-ATPase and ATG16L1. Strikingly, parasites exhibited reduced survival in cell lines deficient in CASM[11], suggesting that this process might serve critical, beneficial functions for the parasite beyond its established role in autophagy.

TFEB belongs to the microphthalmia-associated transcription factor/transcription factor E (MiTF/TFE) family, which orchestrates the expression of the Coordinated Lysosomal Expression and Regulation (CLEAR) gene network[12]. This network, comprising nearly 500 genes identified as direct targets of TFEB, plays a pivotal role in lysosomal biogenesis and the maintenance of cellular energy homeostasis. Lysosomes are organelles that break down waste materials and cellular debris, and their biogenesis is crucial for facilitating all autophagic processes, encompassing cargo recognition, autophagosome formation, and the subsequent fusion of autophagosomes with lysosomes[13].

Activation of TFEB includes shuttling between cytoplasm, the surface of lysosomes, and the cell nucleus, which is primarily governed by phosphorylation state. This is intricately linked to the lysosomal nutrient-sensing mechanism, where the surface of lysosomes acts as a hub for the activation of the mechanistic target of rapamycin complex 1 (mTORC1). mTORC1 responds to cellular nutrient levels, regulating the lysosomal network by modulating the activity of the MiTF/TFE transcription factor family, including TFEB. In nutrient-rich conditions, mTOR and other kinases such as ERK, GSK3, and AKT phosphorylates TFEB, maintaining it in an inactive state in the cytosol[14-16]. Under conditions of starvation, mTORC1 is inactive, allowing TFEB to translocate to the nucleus and activate the transcription of its target genes. Upon restoration of cellular nutrients, TFEB activity is quickly downregulated, and it is exported from the nucleus, ceasing the transcriptional activation of its targets[17,18]. Interestingly, the mechanisms by which mTORC1 regulates TFEB phosphorylation differ from those of other mTORC1 substrates, such as S6K and 4E-BP1. These differences define the canonical and non-canonical branches of mTORC1 signaling. Canonical mTORC1 signaling typically involves a broad, non-selective regulation of multiple substrates, whereas non-canonical signaling is more substrate-specific and involves distinct regulatory mechanisms. For example, in the case of TFEB, non-canonical mTORC1 signaling is regulated by the Folliculin (FLCN) protein, a tumor suppressor that functions as a GTPase-activating protein (GAP) for the RagC/D GTPases. FLCN activity is critical in determining whether TFEB is phosphorylated and retained in the cytoplasm or activated and translocated to the nucleus[19,20].

A recent study has unveiled a novel activation mechanism for TFEB that is not reliant on canonical mTORC1 signaling. This mechanism involves the lipidation of GABARAP family proteins through CASM, leading to the sequestration of the FLCN-FNIP tumor suppressor complex and selective activation of the TFEB pathway[21]. The GAP function of FLCN-FNIP normally promotes the GDP-bound state of the lysosomal Rag GTPases RagC/D, which is permissive for physical binding to TFEB. This then allows optimal presentation of TFEB to mTORC1 resulting in direct phosphorylation of TFEB and its retention in the cytoplasm[20]. However, when FLCN-FNIP is sequestered via CASM, the RagC/D proteins exist in the GTP-bound state, which is not permissive for TFEB binding or mTORC1-dependent phosphorylation. Thus, CASM is a unique mechanism to selectively activate TFEB while preserving canonical mTORC1-dependent signaling pathways such as protein translation[21]. In relation to infection, this unique pathway was first identified in a *Salmonella* infection model, where GABARAPs targeting the *Salmonella*-containing vacuoles (SCVs) effectively sequestered FLCN, leading to the activation of TFEB,

which is considered inducing catabolic activities. The involvement of CASM in this process was confirmed as it was shown to be susceptible to inhibition by SopF and Bafilomycin A1, both inhibitors of the V-ATPase[21]. Thus, CASM-mediated sequestration of FLCN-FNIP on the lysosmal surface of pathogen-containing vacuoles results in TFEB activation while simultaneously maintaining anabolic mTORC1 signaling.

Here we investigate TFEB activation in *P. berghei*-infected cells during the LS development. We have found that proteins of the GABARAP subfamily of ATG8 proteins target the FLCN-FNIP complex to the PVM in response to CASM. Recruitment of FLCN-FNIP results in robust TFEB activation independent of canonical mTORC1 signaling but dependent on CASM and GABARAP proteins. Our work provides mechanistic insight into a novel example of how *P. berghei* manipulates host cell signaling at its PVM.

## Results

### TFEB localizes to the host cell nucleus during *Plasmodium berghei* infection
Under normal conditions, TFEB remains in the cytoplasm of cells, bound to a 14-3-3 protein[14,15] in its phosphorylated form. However, in response to stressors such as starvation or lysosomal damage, TFEB becomes dephosphorylated and translocates to the nucleus, where it activates target genes. To investigate TFEB localization during *P. berghei* infection, we utilized HeLa cells stably expressing TFEB-mCherry (TFEBmCh) and GFP-expressing *P. berghei* parasites (*Pb*GFP). Following infection with *P. berghei* sporozoites, the cells were fixed at various time points for immunofluorescence analysis (IFA). We noted a progressive increase in nuclear localization of TFEB during the course of infection, with peak activation observed 24 h post-infection (hpi) (Fig. 1a, c). Although reduced at 30 and 48 hpi, nuclear TFEB levels remained higher compared to the cytoplasmic levels. Control cells, both in nutrient-rich and starvation media, were also evaluated for comparison, indicating that TFEB activation in starved and *Plasmodium*-infected cells reached similar levels (Fig. 1b, c).

Pivotal for cellular metabolic equilibrium is the kinase activity of mTORC1, favoring anabolism through substrates such as S6K and 4E-BP1 and repressing catabolism by phosphorylating autophagy initiators like ULK1, ATG13, and TFEB[20].

To investigate whether the observed increase in TFEB activation corresponded with a decrease in mTORC1 kinase activity in *P. berghei*-infected cells, we employed phospho-specific antibodies targeting S6 and 4E-BP1. Surprisingly, our results revealed that mTORC1 activity remained consistent with that observed in uninfected control cells, despite concurrent TFEB activation (Figs. 1d, S1). To further assess mTORC1's responsiveness in this context, we exposed both infected and uninfected HeLa cells to conditions of nutrient deprivation and treatment with Torin 1, an inhibitor of mTOR. Under these experimental conditions, we noted a marked suppression of mTORC1 signaling, as evidenced by decreased phosphorylation levels (Figs. 1e, S1). These findings suggest that mTORC1 activity toward TFEB is inhibited within the setting of *P. berghei* infection, thus leading to TFEB nuclear translocation, whereas mTORC1 activity toward canonical substrates S6 and 4EBP1 remained unchanged.

Given that HeLa cells are not the natural host for the parasite, we further validated our findings in primary mouse hepatocytes and Huh7 cells, a human hepatoma cell line commonly used in *Plasmodium* research, observing similar mTOR activity levels in both infected and non-infected cells across these models (Fig. S1). This reinforces the validity of our results and rules out any potential artifacts from using HeLa cells.

The co-activation of anabolic and catabolic pathways in *P. berghei*-infected cells suggests the involvement of the non-canonical branch of mTORC1 signaling[19,20].

### TFEB nuclear translocation depends on CASM and GABARAP proteins
The ATG8 family proteins (LC3A, LC3B, LC3C and GABARAP, GABARAPL1, GABARAPL2) can be conjugated to single membranes in a

process known as CASM, which is reliant on V-ATPase and the ATG5-12-16 complex. We have recently shown that in *P. berghei*-infected cells, LC3B is directly conjugated to the PVM of the parasite in a CASM-like manner[11].

This form of ATG8 conjugation, independent of canonical autophagy, is sufficient for TFEB activation, detaching it from mTORC1 regulation. To verify whether TFEB activation in infected cells is contingent on CASM and

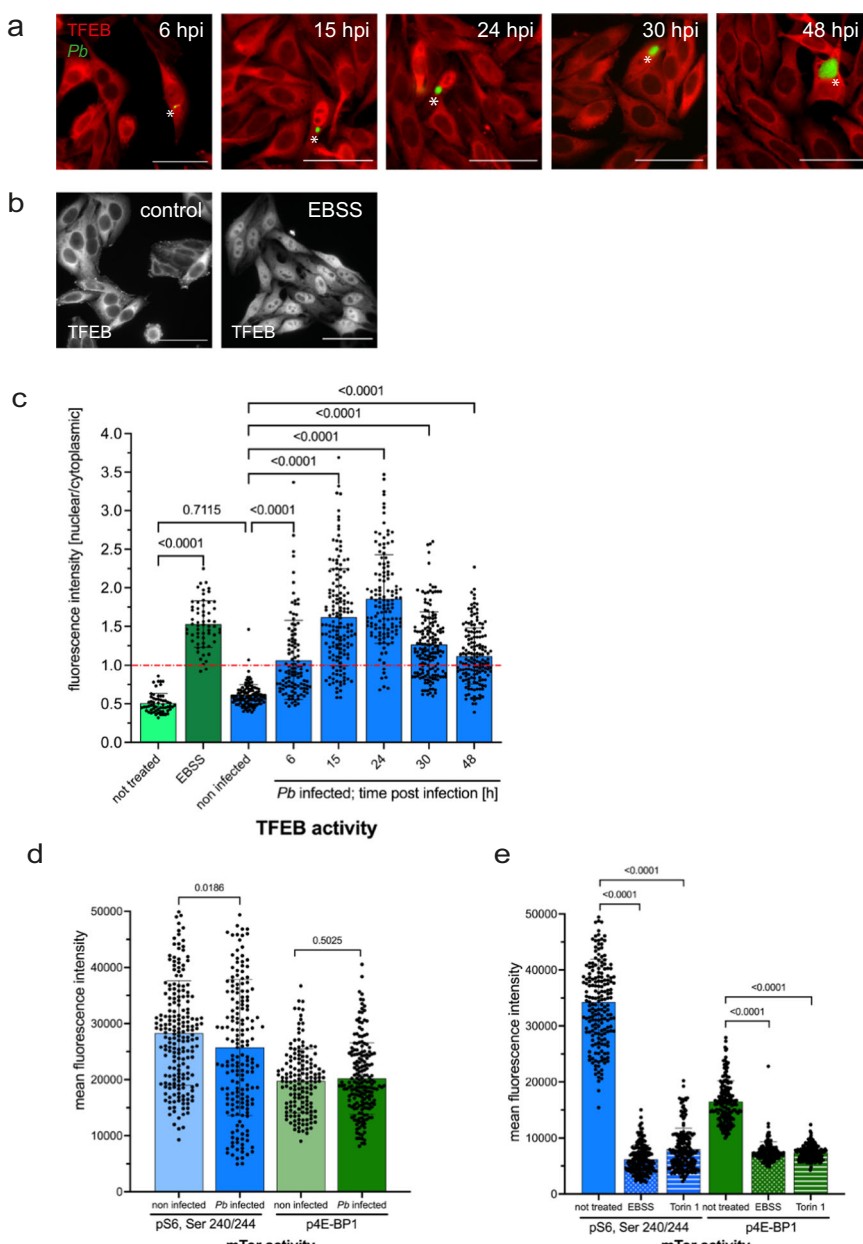

**Fig. 1 | TFEB translocates to the host cell nucleus upon *P. berghei* infection.**
**a** HeLa cells constitutively expressing TFEBmCh were infected with *Pb*GFP sporozoites. Samples were fixed at 6, 15, 24, 30, and 48 hpi and stained with anti-GFP (green) and anti-RFP (red) antibodies to enhance the signals. Samples were analyzed with a widefield fluorescent microscope. Parasites are labeled with a white asterisk. Scale bar 50 μm. Note that in infected cells, TFEBmCh locates to the host cell nucleus whereas in non-infected cells TFEBmCh is in the cytoplasm.
**b** HeLa cells expressing TFEBmCh were grown in full medium (control) or in starvation medium (EBSS) for 2 h. Samples were fixed and stained with anti-RFP antibodies to enhance the TFEB signal here shown in gray. Only in starved cells TFEB localizes to the cell nucleus. **c** Quantification of the experiment described in (**a**) and (**b**). Cells were fixed at indicated time points and stained with anti-GFP (only infected cells) and anti-RFP antibodies, pictures were taken with a widefield fluorescent microscope. Fluorescence intensity in the nucleus and the cell cytoplasm was measured and the ratio nuclear/cytoplasmic was calculated for each cell. A ratio above 1 indicates more nuclear than cytoplasmic TFEB, a ratio lower than 1 indicates more cytoplasmic than nuclear TFEB. Note that

starved and *Pb*-infected cells have a ratio above 1 which means activated TFEB. Pictures were analyzed using Fiji. $N > 30$ for each cell line in each experiment. The graph depicts mean and SD of the pooled data of two independent experiments. *P*-values were calculated using a one-way ANOVA test. **d** mTORC1 activity in *Pb*-infected HeLa cells. Non-infected and *Pb*mCh-infected cells were fixed 24 hpi and stained with anti-pS6(Ser240/244) or with anti-p4E-BP1 antibodies to visualize phosphorylated substrates of the mTOR kinase. Note that mTOR is active in *Pb*-infected cells at the same level as in non-infected cells. Pictures were taken with a widefield fluorescent microscope and fluorescence intensity was measured using Fiji. $N > 60$ for each sample in each experiment. The graph depicts mean and SD of the pooled data of two independent experiments. *P*-values were calculated using a Student's t test. **e** Control experiment for experiment described in (**d**). mTORC1 activity in non-treated, starved (EBSS, 2 h) and Torin 1 (200 nM, 2 h) treated HeLa cells. Cells were fixed and stained with anti-pS6(Ser240/244) or with anti-p4E-BP1 antibodies to visualize phosphorylated substrates of the mTOR kinase. Note that Torin 1 and EBSS inhibit mTOR kinase. Experiment was performed as described in (**d**).

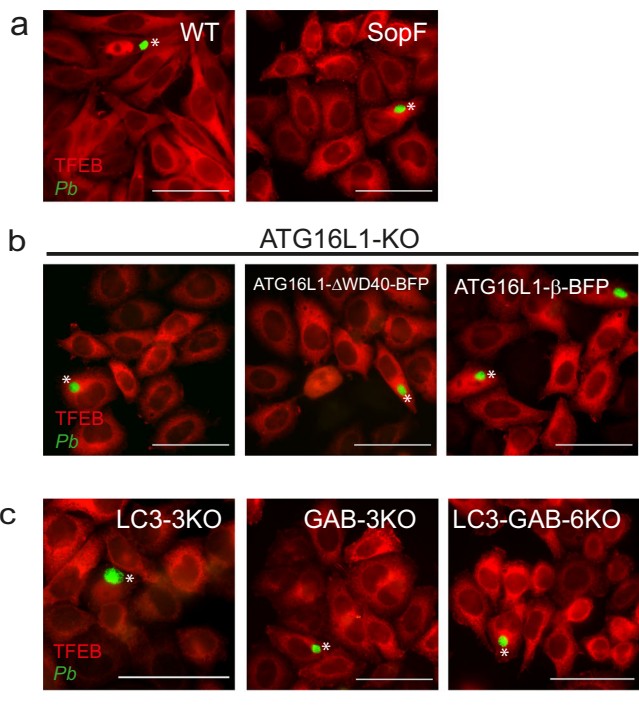

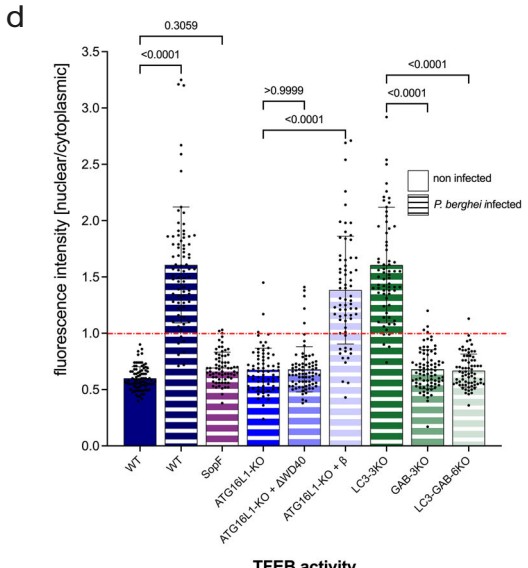

**Fig. 2 | *P. berghei* induced TFEB nuclear translocation depends on CASM and GABARAP proteins. a** SopF inhibits TFEB nuclear translocation in *Pb*-infected HeLa cells. TFEBmCh, SopF expressing HeLa cells were infected with *Pb*GFP sporozoites. 24 hpi cells were fixed and stained with anti-GFP (green) and anti-RFP (red) antibodies to enhance the signals. Samples were analyzed with a widefield fluorescence microscope. Parasites are labeled with a white asterisk. Scale bar 50 μm. Note, in *Pb*-infected WT cells TFEB always localizes to the host cell nucleus, whereas in SopF expressing cells TFEB is cytoplasmic, even in infected cells. **b** *Pb* induced TFEB nuclear translocation depends on ATG16L1-WD40 domain. HeLa cells lacking ATG16L1, reconstituted with either ATG16L1-ΔWD40-BFP or with ATG16L1-β-BFP all expressing TFEBmCh, were infected with *Pb*GFP sporozoites. 24 hpi cells were treated as described in (**a**). Parasites are labeled with a white asterisk. Scale bar 50 μm. Note that nuclear TFEB can be found only in ATG16L1-KO cells expressing the WT ATG16L1-β-BFP. In ATG16L1-KO cells or KO cells reconstituted with ATG16L1-ΔWD40-BFP TFEB is trapped in the cytoplasm. **c** GABARAPs are indispensable for *Pb* induced TFEB nuclear translocation. TFEBmCh expressing HeLa cells lacking all 3 LC3s (LC3A, LC3B, LC3C), or all 3 GABARAPs (GAB, GABL1, GABL2) or lacking all LC3s and all GABs were infected with *Pb*GFP. 24 hpi cells were treated as described in (**a**). Parasites are labeled with a white asterisk. Scale bar 50 μm. Note that TFEB nuclear translocation only happens in the cell line expressing GABARAPs. **d** Quantification of the experiments described in (**a**), (**b**), and (**c**). All cell lines constitutively express TFEBmCh and were infected with *Pb*GFP. Cells were fixed 24 hpi and stained with anti-GFP (only infected cells) and anti-RFP antibodies, pictures were taken with a widefield fluorescence microscope. Fluorescence intensity in the nucleus and the cell cytoplasm was measured and the ratio nuclear/cytoplasmic was calculated for each cell A ratio above 1 indicates more nuclear than cytoplasmic TFEB, a ratio lower than 1 indicates more cytoplasmic than nuclear TFEB. Pictures were analyzed using Fiji. $N > 30$ for each cell line in each experiment. The graph depicts mean and SD of the pooled data of two independent experiments. *P*-values were calculated using a one-way ANOVA test. Note that *Pb* induced TFEB nuclear translocation depends on the ATG16L1-WD40 domain and on GABARAPs and can be inhibited by SopF.

on potential functional redundancies. These included cell lines lacking all three LC3 proteins (LC3-3KO), all three GABARAP proteins (GAB-3KO), and all six ATG8 proteins combined (LC3-GAB-6KO). Each of these cell lines expressed a TFEB-mCherry fusion protein to visually track TFEB activation via its localization. Our observations revealed that TFEB activation relies predominantly on the presence of GABARAP proteins. In the LC3-3KO cells, where only LC3 proteins were absent, TFEB was still able to translocate to the nucleus. However, in the GAB-3KO cells, which lacked all GABARAP proteins, TFEB remained in the cytoplasm, indicating a lack of activation. This differential localization underscores the essential role of GABARAP proteins in facilitating TFEB activation, as opposed to LC3 proteins, which appear not to be necessary for this process (Fig. 2c, d). Notably, all cell lines employed in this study were capable of inducing TFEB nuclear translocation in response to nutrient deprivation, a process typically regulated by mTORC1 through canonical autophagy pathways (Fig. S2a–d). This suggests that TFEB activation in *P. berghei*-infected cells is mediated by a substrate-specific, non-canonical mTORC1 signaling pathway that specifically depends on GABARAP ATG8 proteins and the CASM pathway.

## GABARAP proteins localize to the *P. berghei* PVM depending on ATG16L1

In our study, we observed that TFEB activation in *P. berghei*-infected cells depends on GABARAP proteins (Fig. 2c, d). This led us to investigate the precise localization of these proteins. We have shown previously that LC3B is directly incorporated into the parasite's PVM via a mechanism similar to CASM[11]. Based on this, we hypothesized that GABARAP proteins might also integrate into the PVM in a similar fashion. To test this, we transfected HeLa WT, ATG16L1-KO, or SopF-expressing cells with plasmids expressing each of the GABARAP proteins fused to GFP. Subsequently, we infected these cells with *Pb*mCh sporozoites, fixed them 6 h post-infection (hpi), and conducted IFA using anti-GFP antibodies to enhance the signal. Our results clearly demonstrate that all three GABARAPs localize at the

GABARAP proteins, we examined TFEB localization in cells expressing SopF, ATG16L1 knockout (KO) cells, and various ATG8 KO cell lines, all stably expressing TFEBmCherry (TFEBmCh). In HeLa WT cells, TFEB predominantly localized to the nucleus during infection. In contrast, in cells expressing the CASM inhibitor SopF, a *Salmonella typhimurium* protein that disrupts CASM by targeting the V0 complex of the V-ATPase, TFEB remained cytoplasmic (Fig. 2a, d). This underscores the critical role of V-ATPase-ATG16L1 interaction in ATG8 lipidation and subsequent TFEB activation. In ATG16L1 deficient cells, or those expressing a truncated form of ATG16L1 lacking the WD40 domain required for CASM, TFEB failed to translocate to the nucleus, highlighting the domain's essential role in this process. However, reintroduction of ATG16L1 in KO cells restored TFEB activation by the parasite (Fig. 2b, d).

To thoroughly examine the role of ATG8 protein subfamilies in the activation of TFEB, we employed cell lines with specific knockouts for clarity

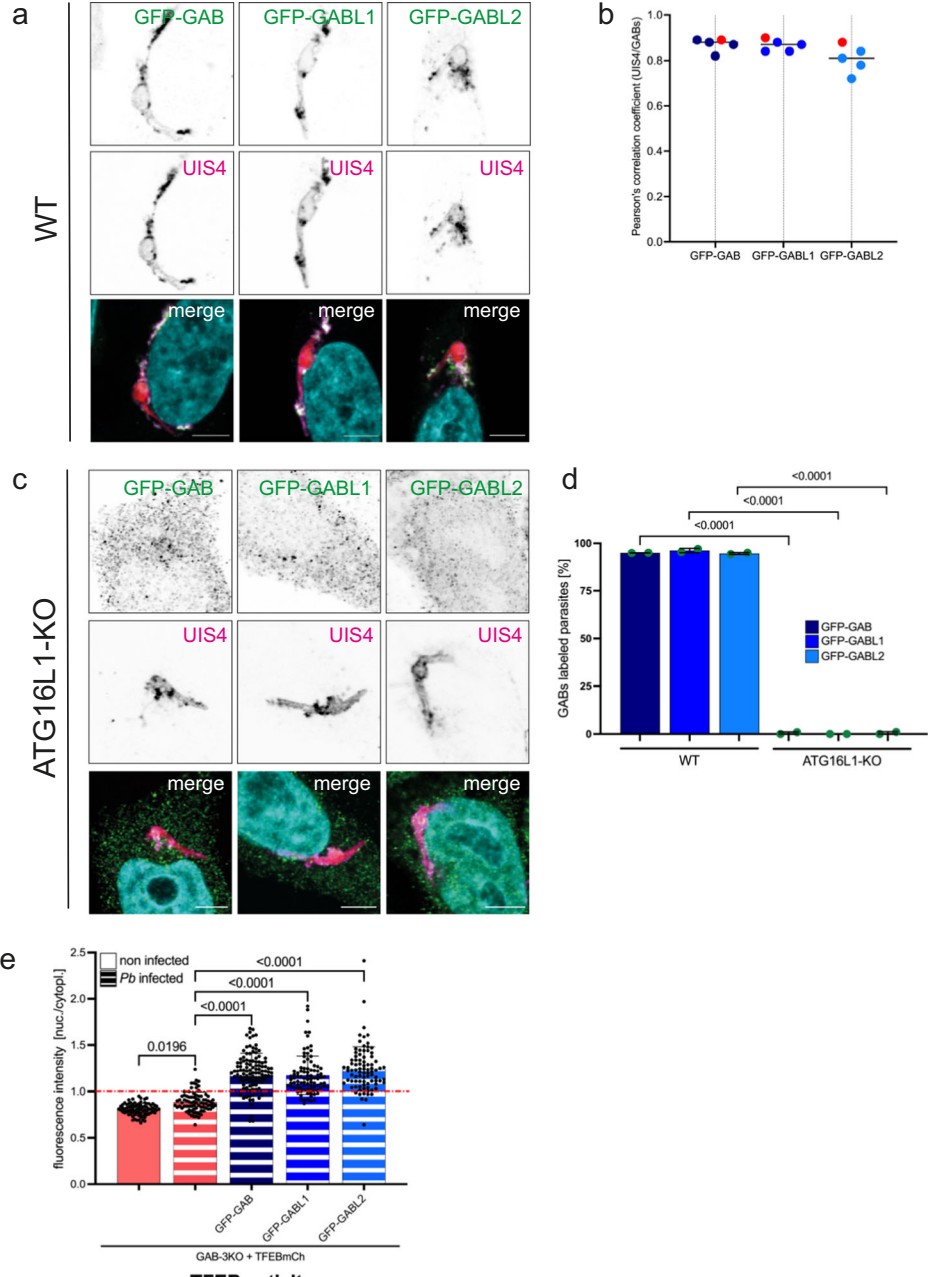

**Fig. 3 | GABARAPs localize to *Plasmodium* PVM depending on ATG16L1. a** All 3 GABARAP proteins localize to *Pb* PVM. HeLa WT cells were transiently transfected with GFP-GABARAPs and approximately 15 h post transfection infected with *Pb*mCh sporozoites. 6 hpi infected cells were fixed and stained with anti-GFP antibodies (green) to enhance the GABARAP signal and anti-UIS4 antibodies (magenta) to visualize the *Pb* PVM. DNA was stained with Dapi (cyan). Images were taken with a confocal laser scanning microscope. Scale bar 5 µm. Note that GABARAP, GABARAPL1, and GABARAPL2 clearly localize to the *P. berghei* PVM. **b** Quantification of (**a**). Graph shows the Pearson's correlation coefficient (PCC) for UIS4 and GFP-GABARAPs. PCC was calculated using the Coloc2 tool of FIJI. *N* = 5 parasites. Each dot represents one parasite, each red dot represents the parasites shown in (**a**). **c** PVM localization of GABARAPs depends on ATG16L1. HeLa cells lacking ATG16L1 were transiently transfected with GFP-GABARAPs and treated as described in (**a**). Images were taken with a confocal laser scanning microscope. Scale bar 5 µm. Note that none of the GABARAP proteins localizes to the *Pb* PVM in ATG16L1-KO cells. **d** Quantification of the experiments described in (**a**) and (**c**).

The graph shows the percentage of GABARAP-positive parasites in HeLa WT and ATG16L1-KO cells. Only UIS4-positive parasites were counted. The graph depicts the mean and SD of two independent experiments. *P*-values were calculated using a Student's t test. $N \geq 70$ per experiment and cell line. **e** GABARAPs are needed for nuclear translocation of TFEB in *Pb*-infected cells. Quantification of the TFEB signal in GAB-3KO cells and GAB-3KO cells transiently transfected with each of the GFP-GABARAPs all constitutively expressing TFEBmCh. Cells were fixed 24 hpi and stained with anti-GFP and anti-RFP antibodies, pictures were taken with a widefield fluorescence microscope. Fluorescence intensity in the nucleus and the cell cytoplasm was measured and the ratio nuclear/cytoplasmic was calculated for each cell. A ratio above 1 indicates more nuclear than cytoplasmic TFEB, a ratio lower than 1 indicates more cytoplasmic than nuclear TFEB. Note that all GABARAPs are proficient to activate TFEB upon *Pb* infection. Pictures were analyzed using Fiji. *N* > 30 for each cell line in each experiment. The graph depicts mean and SD of the pooled data of two independent experiments. *P*-values were calculated using a one-way ANOVA test.

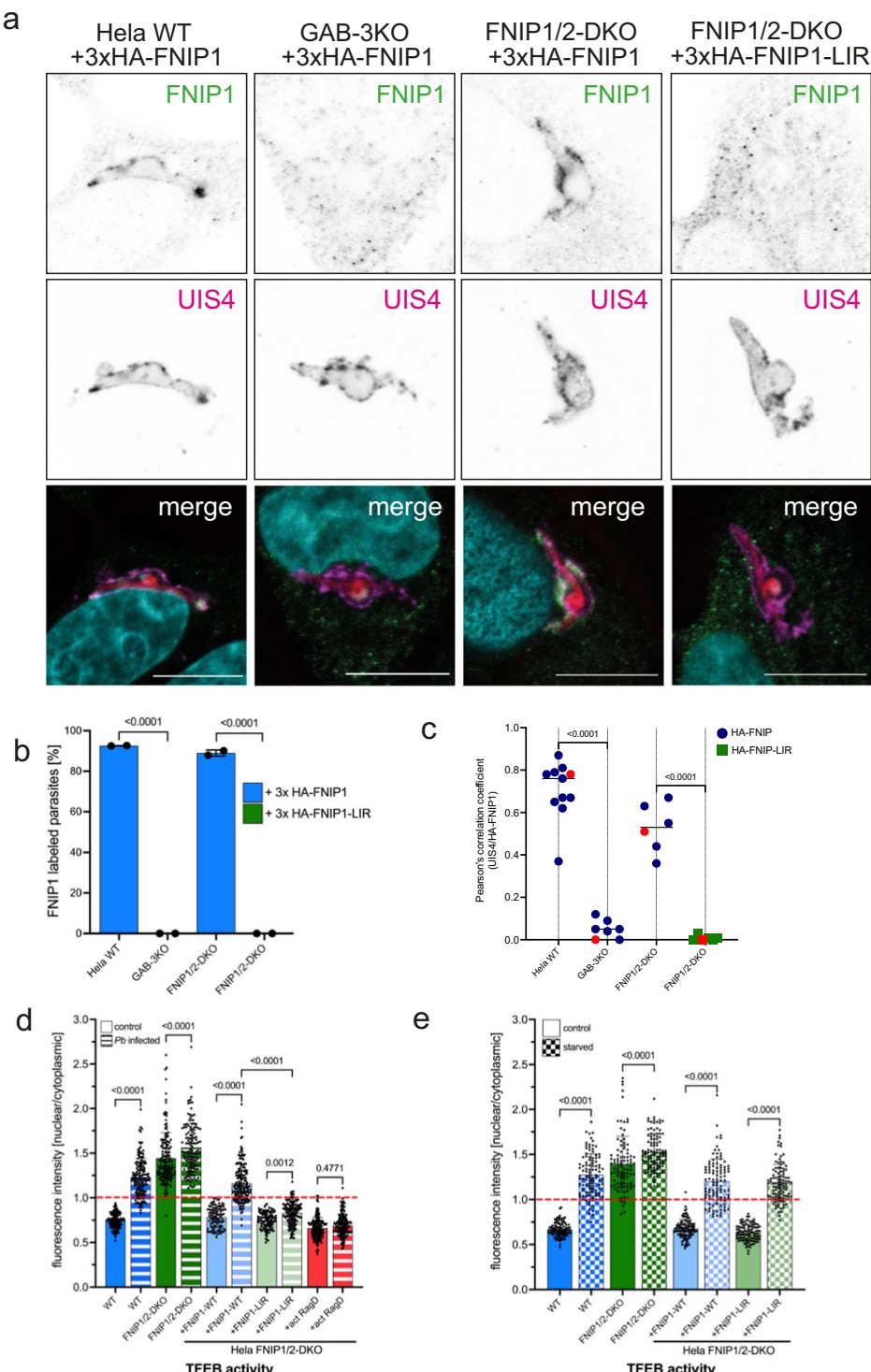

PVM of *P. berghei*, as confirmed by antibody staining of UIS4, a marker protein for the parasite PVM (Fig. 3a, b, d). Further analysis, calculating the Pearson's correlation coefficient between GABARAPs and UIS4, revealed values higher than 0.8, confirming the close proximity of the two proteins (Fig. 3b). More than 94% of the parasites exhibited GABARAP decoration, and all three GABARAPs equally localized to the parasite's PVM 6 hpi (Fig. 3d). Notably, ATG16L1 knockout cells and cells expressing the bacterial protein SopF, both deficient in CASM, showed no localization of GABARAP proteins at the parasite PVM (Figs. 3c, d, S3a, b).

In our experimental setup, we employed ectopically expressed GFP-GABARAP constructs in HeLa cells. To rule out the possibility of observed

PVM localization being an artifact of ectopic overexpression or the use of GFP fusion proteins, we extended our analysis to investigate the localization of endogenous GABARAPs not only in HeLa cells but also in Huh7 cells and primary mouse hepatocytes. Utilizing a newly developed target-binding reagent called Ankyron, we confirmed the endogenous GABARAP localization at the *P. berghei* PVM (Fig. S4a–c). Ankyrons, small ankyrin repeat-based proteins, specifically bind their target protein with high affinity. The Ankyron used in our study binds to all three GABARAP proteins, with GABARAP displaying the strongest labeling. In addition, we observed nonspecific nuclear staining, which was also present in non-transfected GAB-3KO cells (Fig. S4d).

**Fig. 4 | FNIP1 localizes to *Pb* PVM depending on GABARAPs and on FNIP1-LIR domain. a** FNIP1 localization at *P. berghei* PVM. HeLa WT, GAB3-KO, and FNIP1/2-DKO cells all constitutively expressing 3x-HA-FNIP1 were infected with *Pb*mCh. 6 hpi cells were fixed and stained with anti-HA (green) and anti-UIS4 (magenta). *Pb*mCh is shown in red. DNA was stained with Dapi (cyan). Images were taken with a confocal laser scanning microscope. Scale bar 10 μm. Note that the PVM localization of FNIP1 is dependent on GABARAPs and on a functional FNIP1-LIR domain. **b** Quantification of the experiment described in (**a**). The graph shows the percentage of FNIP1-positive parasites in the different HeLa cell lines. Only UIS4-positive parasites were counted. The graph depicts the mean and SD of two independent experiments. *P*-values were calculated using a Student's t test. $N \geq 70$ per experiment and cell line. **c** Quantification of the experiment described in (**a**). Graph shows the Pearson's correlation coefficient (PCC) for UIS4 and 3xHA-FNIP1. PCC was calculated using the Coloc2 tool of FIJI. $N = 11$ parasites for HeLa WT, 7 parasites for GAB-3KO, 6 for FNIP1/2-DKO + 3xHA-FNIP1, and 6 parasites for FNIP1/2-DKO + 3xHA-FNIP1-LIR. Each symbol represents one parasite, each red symbol represents the parasites shown in (**a**). **d** TFEB activation in infected cells depends on *Pb* PVM localization of FNIP1. All cell lines constitutively express TFEBmCh. FNIP1-DKO cells additionally express 3xHA-FNIP1, 3xHA-FNIP1-LIR

or are transiently transfected with HA-RagD77L (a constitutively active mutant of RagD), as indicated. Cells were infected with *Pb*mCh and fixed 24 hpi and stained with anti-RFP and anti-HA (only RagD transfected). Pictures were taken with a widefield fluorescence microscope. Fluorescence intensity in the nucleus and the cell cytoplasm was measured and the ratio nuclear/cytoplasmic was calculated for each infected cell. A ratio above 1 indicates that more TFEB is in the nucleus than in the cytoplasm and a ratio lower than 1 indicates mor TFEB in the cytoplasm than in the nucleus. Pictures were analyzed using Fiji. $N > 50$ for each cell line in each experiment. The graph depicts mean and SD of the pooled data of two independent experiments. Single-color bars represent non-infected cells, striped bars represent *Plasmodium*-infected cells. *P*-values were calculated using a one-way ANOVA test. Note that TFEB is only activated when FNIP1 is sequestered to the *Pb* PVM. **e** Starvation control for the cell lines used in the experiment described in (**d**). Cells were left untreated (single-color bars) or starved for 2 h in EBSS (dotted bars). The analysis was carried out as described in (**d**). $N > 50$ for each cell line in each experiment. The graph depicts mean and SD of the pooled data of two independent experiments. *P*-values were calculated using a one-way ANOVA test. Note that a functional FNIP1 LIR domain is not needed for TFEB activation in starved cells.

---

Our investigation revealed that GABARAP proteins are essential for TFEB activation in *P. berghei*-infected cells (Fig. 2c, d). This prompted us to inquire whether each of the three GABARAPs could independently activate TFEB or if they possess distinct functions in infected cells. To address this, we separately transfected GAB-3KO cells expressing TFEBmCh with each GFP-GABARAP construct. Subsequently, we infected these cells with *P. berghei* sporozoites 15 h post-transfection and analyzed TFEB activation 24 hpi. Remarkably, each individual GABARAP alone was proficient at promoting TFEB nuclear localization, suggesting redundant functions among the three GABARAP proteins (Figs. 3e, S5).

## FNIP1 is sequestered at the *P. berghei* PVM via its interaction with GABARAPs

To further test our hypothesis that TFEB is activated through non-canonical mTORC1 signaling, we next sought to determine FNIP1 localization in *Plasmodium*-infected cells. Under typical conditions, the FNIP-FLCN complex functions as a GAP for RagC/D with GDP-bound RagC/D promoting the interaction with TFEB at the lysosome surface. Consequently, TFEB comes into close proximity to mTORC1, undergoes phosphorylation, and is retained in the cytoplasm. Goodwin et al. demonstrated that FNIP1 directly interacts with GABARAP, but not LC3, via a recently described LIR motif (LC3-interacting motif). Based on these findings, we anticipated observing FNIP1 localizing to the parasite's PVM.

Both HeLa WT and GAB-3KO cells, constitutively expressing 3xHA-FNIP1, were infected with *Pb*mCh sporozoites, fixed 6 h post-infection (hpi), and subjected to IFA using anti-HA and anti-UIS4 antibodies. FNIP1 was found to localize at the PVM of *P. berghei* parasites in cells expressing GABARAP proteins (Fig. 4a–c).

To delve deeper into the functional dynamics of the FNIP1 and FNIP2 proteins, particularly their interactions with GABARAP, we have employed FNIP1/2 double knockout (DKO) cell lines. These cell lines are specifically designed to lack both FNIP1 and FNIP2, enabling a clearer understanding of their roles in cellular processes without the confounding effects of compensatory mechanisms that often occur when only one gene is knocked out. Importantly, FNIP2, like FNIP1, forms part of the FNIP-FLCN complex, which is involved in various cellular regulatory mechanisms including autophagy and energy metabolism[22].

In our studies, we used these FNIP1/2-DKO cell lines to assess the significance of the recently characterized LIR motif in FNIP1 for its interaction with GABARAP proteins. By expressing a mutated version of FNIP1 with a disrupted LIR motif (Y583A, V586A, FNIP1-LIR) in these cells, we aimed to determine whether this specific motif is essential for the localization of FNIP1 to the PVM of the parasite. In FNIP1/2-DKO cells expressing the wild-type FNIP1, the protein successfully localized to the parasite's PVM, confirming the functionality of the FNIP1-GABARAP interaction.

Conversely, in FNIP1/2-DKO cells expressing the LIR mutant, FNIP1 was not detected at the parasite's PVM (Fig. 4a–c). This contrast underscores the critical role of the LIR motif in facilitating the association between FNIP1 and GABARAPs at the PVM.

FNIP1 forms a heterodimeric complex with FLCN, which we aimed to visualize in the context of *P. berghei* infection. For this purpose, we transfected cell lines stably expressing 3xHA-FNIP1 with GFP-FLCN. In WT cells or FNIP1/2-DKO cells expressing 3xHA-FNIP1, FLCN localized to the parasite PVM alongside FNIP1 (Fig. S6). However, in HeLa cells lacking GABARAP proteins (GAB-3KO) or when the LIR motif of FNIP1 is mutated (FNIP1/2-DKO with 3xHA-FNIP1-LIR), FLCN could not be detected at the parasite's PVM (Fig. S6). This demonstrates that FNIP1 recruits FLCN to the PVM and that FLCN alone does not bind to GABARAPs or any other PVM protein. These findings further validate the utility of FNIP1/2-DKO cell lines in dissecting the molecular interactions within the FNIP-FLCN complex at the PVM.

## The interaction between FNIP and GABARAP leads to TFEB activation in *P. berghei*-infected cells via the non-canonical mTORC1 signaling pathway

In the non-canonical mTORC1 pathway, the FLCN-FNIP complex exhibits GAP activity toward RagC/D. RagC/D is active when bound to GDP, allowing TFEB to bind and undergo phosphorylation by mTORC1. Cells lacking FNIP1/2 cannot activate RagC/D, resulting in consistently non-phosphorylated nuclear TFEB[19].

In *Plasmodium*-infected cells, TFEB is activated and localizes to the host cell nucleus (Fig. 1a, c). To determine whether this TFEB activation relies on the FNIP1:GABARAP protein interaction, we infected various cell lines, all stably expressing TFEBmCh, and analyzed TFEB localization 24 hpi. In FNIP1/2-DKO cells where the FLCN-FNIP complex cannot form, TFEB is localized in the cell nucleus, regardless of infection status, as expected (Figs. 4d, S7a). FNIP1/2-DKO cells expressing 3xHA-FNIP1 exhibit nuclear TFEB only when infected with *Plasmodium* (Figs. 4d, S7a). In contrast, cells expressing FNIP1 with a mutated LIR motif (3xHA-FNIP1-LIR), which cannot interact with GABARAPs and thus cannot be sequestered at the PVM, rarely exhibit more nuclear than cytoplasmic TFEB, even in *P. berghei*-infected cells (Figs. 4d, S7a). FNIP-LIR, lacking GABARAP interaction, remains free from sequestration by the parasite, allowing it to activate RagC/D and resulting in phosphorylated cytoplasmic TFEB. Similar results are observed when a constitutive active RagD mutant is expressed in FNIP1/2-DKO cells, leading to no TFEB activation in *P. berghei*-infected cells (Figs. 4d, S7b). It appears that the parasite sequesters FNIP1 at its PVM through interaction with GABARAPs, rendering FNIP1 unavailable as a RagC/D activator at the lysosomal surface. As a proof of principle, we incubated FNIP1/2-DKO cells expressing either 3xHA-FNIP1

**Fig. 5 | TFEB supports parasite survival. a** *P. berghei* survival is reduced in TFEB-KO cells. HeLa WT, TFEB-KO, and TFEB-KO + TFEBmCh cells were infected with *PbGFP*. At 6 and 48 hpi, parasite numbers were evaluated using automated high throughput live cell imaging and analysis (INCell Analyzer 2000). The graph shows relative parasite survival from 6 to 48 hpi compared to the WT control. The mean and SD of three independent experiments are depicted. *N* > 500 parasites per cell line and experiment. A one-way ANOVA test was used to determine *p*-values. **b** Parasite size at 48 hpi of the experiment described in (**a**). The experiment was performed three times. Shown is the data of one representative experiment. Median and SD are depicted. *P*-values were calculated using a one-way ANOVA test. HeLa WT *N* = 193; TFEB-KO *N* = 129; TFEB-KO + TFEBmCh *N* = 237. **c** Western blot of the cell lines used in (**a**) and (**b**). Whole protein lysates were separated on a 12% acrylamide gel, transferred onto a nitrocellulose membrane, and probed with an anti-TFEB antibody. α-tubulin was detected as a loading control. Note that in line 3, the TFEBmCh fusion protein expressed in the TFEB-KO cells is larger than the endogenously expressed TFEB due to the mCherry fusion partner (see also Fig. S8a, b).

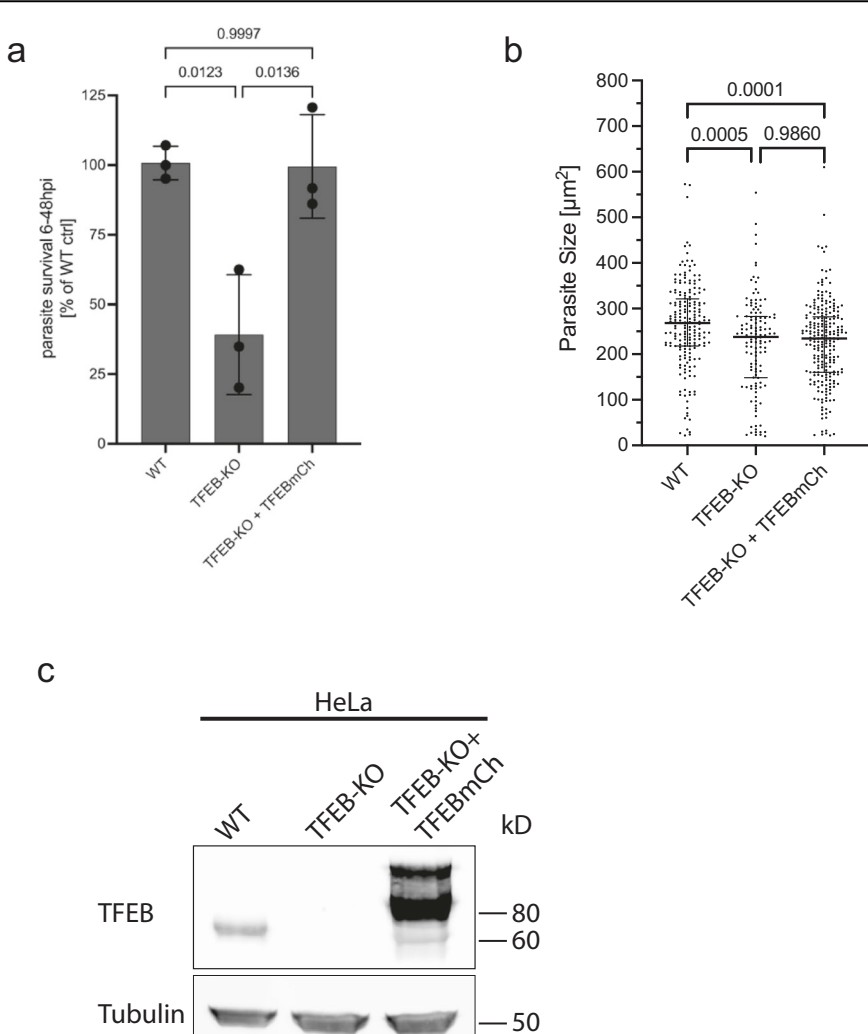

or 3xHA-FNIP1-LIR in EBSS. Both cell lines were able to activate TFEB upon starvation, regulated by the canonical mTOR signaling pathway (Figs. 4e, S7c). This demonstrates the robustness of the FNIP1/2-DKO cell model in studying the regulatory mechanisms governing TFEB activation, independent of FNIP1's interaction with GABARAP through the LIR motif.

**TFEB signaling supports parasite survival**

Finally, we wanted to know whether TFEB activation has any physiological relevance for parasite liver stage development. Through automated high-throughput live cell imaging, we analyzed parasite development in TFEB-KO cells, monitoring parasite survival from 6 to 48 hpi and assessing parasite size at 48 hpi. Strikingly, we observed a significant reduction in parasite survival in TFEB-KO cells to less than 50% compared to parasites infecting WT cells (Fig. 5a). Interestingly, parasite size was significantly reduced in TFEB-KO cells, although the size reduction was modest (268 μm² vs. 238 μm²) (Fig. 5b). To confirm the specificity of this phenotype, reconstitution experiments were conducted. Complementation with TFEBmCh fully rescued the drop in survival observed in the TFEB-KO cell line, whereas the decrease in parasite size was not rescued (Fig. 5a, b). These findings indicate that TFEB strongly supports parasite survival and may additionally play a minor role in promoting parasite growth. In fact, TFEB orchestrates the expression of numerous genes involved in lysosomal biogenesis and autophagy, such as LAMP1, LAMP2, and TFE3, as well as genes that enhance cellular catabolism, like those involved in lipid metabolism and the breakdown of proteins and complex carbohydrates. By upregulating these genes, TFEB activation can provide essential resources for the parasite, facilitating its adaptation and persistence in the host environment. This

adaptive mechanism likely contributes to an enhanced resilience and metabolic efficiency of the parasite, thereby supporting its survival and possibly influencing its growth within the host cells.

**Discussion**

*Plasmodium* parasites encounter numerous challenges during their life cycle, particularly during the liver stage, where they proliferate rapidly. These challenges include acquiring sufficient nutrients, evading the host cell's intracellular immune response, and inhibiting host cell death to ensure parasite survival. *Plasmodium* parasites overcome these obstacles by actively manipulating host cell processes such as apoptosis and interfering with host cell signaling pathways[23,24]. In this study, we present a molecular model elucidating how *Plasmodium* parasites activate the pro-survival transcription factor TFEB during liver stage infection (Fig. 6).

TFEB, known as a regulator of autophagy and lysosomal biogenesis, reacts to the cellular nutritional state[25]. However, for intracellular parasites, TFEB activation presents a double-edged sword. On one hand, it enhances the host cell's ability to generate nutrients through the upregulation of genes involved in lysosomal biogenesis, autophagy, and cellular catabolism, providing essential resources that can benefit the parasite. This activation could facilitate the parasite's adaptation, proliferation, and survival by improving the metabolic efficiency within the host environment.

On the other hand, increased autophagy and lysosomal capacity also bolster the host cell's intracellular immune response against pathogens. Autophagy, a critical process in cellular defense, can target and degrade intracellular parasites, thereby limiting their ability to replicate and persist within the host. Similarly, enhanced lysosomal activity may lead to more

## Non-canonical mTORC1 signaling in *Plasmodium berghei* infected cells

**Fig. 6 | Schematic overview of transcription factor EB activation in *Plasmodium berghei*-infected cells.** Normal conditions: active mTORC1 is localized to lysosomes by binding to RagA/B, which in turn binds to the Ragulator complex. This complex interacts with the V-ATPase located in the lysosomal membrane. The FLCN-FNIP1/2 complex, acting as a GTPase-activating protein (GAP) for RagC/D, enables active RagC/D (bound to GDP) to directly interact with TFEB. This interaction positions TFEB near mTORC1, leading to TFEB phosphorylation. Phosphorylated TFEB is then sequestered in the cytoplasm by its binding to a 14-3-3 protein. Active mTORC1 also phosphorylates numerous other substrates, such as 4E-BP1 and S6K, promoting anabolic processes like protein, nucleotide, and lipid synthesis while inhibiting catabolic processes like autophagy and lysosome biogenesis. *P. berghei*-induced TFEB activation: host cell GABARAP proteins are integrated into the parasitophorous vacuolar membrane (PVM) of the parasite shortly after infection, via a mechanism involving V-ATPase-ATG16L1-ATG5-ATG12 (CASM). This leads to the sequestration of the FLCN/FNIP complex at the PVM through interactions with GABARAPs, disrupting its GAP activity. Consequently, RagC/D becomes inactive (bound to GTP), preventing its interaction with TFEB. Without this interaction, TFEB remains unphosphorylated and moves into the nucleus to activate target genes. Meanwhile, mTORC1 remains active and bound to RagA/B, capable of phosphorylating its substrates. In cells infected by *P. berghei*, we observe a unique scenario where both anabolic and catabolic processes are simultaneously activated within the same cell. Schematic created with BioRender.com.

efficient degradation of parasitic components that are sequestered within these organelles. Consequently, while TFEB activation can provide short-term benefits by supplying the parasite with necessary nutrients, it also enhances the host cell's defense mechanisms, potentially leading to the clearance of the parasite over time. This intricate balance makes the regulation of TFEB a critical factor in the survival and pathogenicity of intracellular parasites.[9,26].

Indeed, TFEB exhibits diverging roles in pathogen-infected cells, with its activity manipulated by pathogens (reviewed in ref. [27]). For instance, *Salmonella* bacteria secrete the effector protein SopF to disrupt the V-ATPase-ATG16L1-axis, inhibiting nuclear translocation of TFEB and TFEB-mediated xenophagy[28,29]. Conversely, the *Legionella* effector SetA glycosylates TFEB to inhibit its phosphorylation, resulting in constitutive nuclear localization[30]. This persistent nuclear localization is continuously driving the expression of genes that enhance lysosomal biogenesis and autophagy. For *Legionella*, this manipulation proves advantageous as it creates a nutrient-rich environment by increasing the degradation of cellular components, which the bacteria can utilize for growth. In addition, by promoting a continuous state of lysosomal and autophagic flux, *Legionella* may modify these processes to its benefit, potentially undermining the host cell's ability to use them effectively as a defense mechanism against the pathogen. This strategic alteration of host cell signaling allows *Legionella* to sustain a favorable niche within its host. These examples underscore the versatility of TFEB regulation by intracellular pathogens.

TFEB is known to be regulated through phosphorylation by mTORC1, depending on the nutritional state of the cell. In fed cells, mTORC1 is in its active state, localized at the lysosome, and phosphorylates its substrates, such as TFEB. In starved cells, mTORC1 dissociates from the lysosome and becomes inactive. TFEB remains unphosphorylated and translocates to the nucleus, where it activates its target genes[13]. At the beginning of our study, we were faced with the observation of nuclear TFEB and active mTORC1 simultaneously in *Plasmodium*-infected cells. This implies that both cell anabolism and catabolism are activated concurrently. Napolitano et al. (2020) described that TFEB is regulated through a non-canonical substrate-specific mTORC1 signaling. This pathway is inhibited by the lack of GAP activity of the FNIP-FLCN complex toward RagC/D, leading to TFEB nuclear translocation and consequent activation. This absence, often referred to as the "missing" GAP activity, allows for continuous inhibition of RagC/D, leading to the activation of TFEB independently of nutritional signals. This process is further enabled by the sequestration of the FNIP-FLCN complex to cellular membranes through interactions with GABARAP proteins[21]. We have now demonstrated that GABARAP proteins are incorporated into the PVM in an ATG16L1-dependent manner. Here, the GABARAPs effectively sequester the FNIP/FLCN complex at the PVM, which prevents it from exerting its GAP function on RagC/D. This sequestration supports the non-canonical activation of TFEB, highlighting a critical mechanism that impacts cellular signaling and pathogen survival.

The relevance of GABARAP proteins for TFEB activation during *Plasmodium* infection is intriguing, as they were described to be involved in other processes during *Plasmodium* liver stage development. *P. berghei* parasites lacking UIS3 (a PVM resident protein) are not able to grow in host cells expressing GABARAPs. Surprisingly, these parasites can proliferate in GAB-3KO cells, indicating a role for GABARAPs in host cell-mediated parasite elimination[31]. Furthermore, expression of the GABL1 mRNA is upregulated in mouse hepatocytes infected with *P. berghei*, as shown in a large study employing single-cell RNA sequencing[32]. Interestingly, expression profiling of the GABL1 gene in mouse livers has revealed that it is upregulated by TFEB, and several putative CLEAR elements, the TFEB binding sites, have been identified upstream of the transcription start site by bioinformatic prediction[33]. Moreover, GABARAP proteins are involved in lysosomal fusion[34], which is intriguing considering that a relevant percentage of *Plasmodium* parasites are eliminated by lysosomal fusion[26].

TFEB activation upon GABARAP-dependent FNIP-FLCN complex sequestration was described by Goodwin et al., who used lysosomal perturbations to induce ionic/pH imbalances. This mechanism of TFEB activation is also relevant for selective autophagy, such as mitophagy or xenophagy. The authors demonstrated that the membrane surrounding the *Salmonella*-containing vacuole (SCV) was quickly labeled with GABARAP proteins activating TFEB signaling. This process depends on the V-ATPase-ATG16L1-axis and CASM, similar to our findings during *P. berghei*

infection. Other processes inducing CASM are described where different stimuli converge to cause ionic/pH imbalances of endosomal vesicles. This raises the hypothesis that a mechanism, potentially the V-ATPase, can sense these imbalances, thereby inducing CASM through the V-ATPase-ATG16L1-axis[10]. How the V-ATPase mechanistically senses such ionic/pH imbalances mechanistically remains unknown. It can be speculated that in *Plasmodium*-infected cells, an ionic imbalance in the PV promotes increased association of the V0 and V1 domains of the V-ATPase at the PVM, inducing CASM followed by TFEB activation. There are multiple reasons and recent discoveries that make such a scenario probable. Firstly, it was discovered that the lysosomal marker protein LAMP1 co-localizes with the PVM[26], indicating that lysosomes, where the V-ATPase localizes, fuse with the PVM. Secondly, tagging of a subunit of the host-cell V-ATPase showed its localization at the PVM[11]. Thirdly, although the V-ATPase is present in the PVM, successfully developing parasites are not acidified[26]. This may be due to non-selective channels in the PVM[35] that enable free diffusion of protons across it, resulting in changes in the proton concentration in the PV and to an ionic/pH imbalance i.e., neutralization of the acidic pH. Altogether, this could explain the activation of the upstream processes such as the initiation of the V-ATPase-ATG16L1-axis that ultimately leads to TFEB activation. What remains unknown, however, is whether the host cell V-ATPase also directly assembles at the PVM with its membrane-bound subunit reaching it by secretory transport, or if it only ends up in the PVM due to fusion of lysosomes. Lastly, one might even speculate that *Plasmodium* transports its own V-ATPase, or only the membrane-bound V0 domain, which could then assemble with host-cell V1 domains at the PVM to induce potentially beneficial downstream processes such as TFEB activation during the liver stage, as it has been shown that the *P. falciparum* V-ATPase localizes to the PVM during the blood stage[36].

TFEB significantly contributes to the survival of parasites in *P. berghei*-infected cells. In cells devoid of TFEB, parasite survival drops to less than 50%, although the size of the parasite is less affected. This suggests that TFEB plays a crucial role in supporting parasite survival but is not involved in their growth. Furthermore, TFEB appears to have varied impacts on the development of intracellular pathogens. For instance, in infections with *Leishmania donovani*, TFEB expression is upregulated, where it negatively affects critical immune functions such as antigen presentation and cytokine secretion. Intriguingly, silencing TFEB in infected cells leads to enhanced parasite clearance, indicating that TFEB's presence benefits the parasite[37]. In contrast, during *Staphylococcus aureus* infections in *C. elegans*, the *C. elegans* TFEB homolog, HLH-30, is rapidly activated and translocates to the nucleus upon infection, where it is essential for inducing the expression of host defense genes, including antimicrobial peptides[38]. Thus, TFEB's role extends beyond lysosomal biogenesis and autophagy; it also appears to play a part in the immune response against pathogens[39].

Recently, our lab has discovered Nuclear factor erythroid-derived 2-related factor 2 (NRF2), a host cell transcription factor, supports the survival of *Plasmodium* parasites during the liver stage[24]. NRF2 becomes activated due to sequestration of its negative regulator KEAP1 to the PVM by the selective autophagy receptor p62. This transcription factor is recognized for its role in responding to oxidative stress, such as the detoxifying reactive oxygen species (ROS), by regulating hundreds of genes with cytoprotective function to enhance host cell survival. Notably, there is an interaction between NRF2 and TFEB. TFEB activates NRF2 by inhibiting the transcription of DCAF11, an E3 ubiquitin ligase that targets NRF2 for degradation. This inhibition is due to a direct interaction between TFEB and a CLEAR sequence motif in the 5′ UTR of the DCAF11 gene. Furthermore, active TFEB increases the phosphorylation level of p62, which is essential for sequestering KEAP1 and thereby activating NRF2. These mechanisms enable TFEB to activate NRF2 by improving its stability[40]. In addition, p62 is phosphorylated by mTORC1[41], which may elucidate the TFEB-mediated rise in phospho-p62 levels, as active TFEB has been demonstrated to augment the activity of mTORC1 through transcriptional regulation of mTORC1 components such as RagC/D[42].

In this study, we show that *Plasmodium* parasites disrupt the host cell's non-canonical mTORC1 signaling to enhance their own survival. We describe the molecular mechanism by which the parasite hijacks the FNIP-FLCN complex through the host cell's GABARAP proteins, which are incorporated into the PVM. This integration triggers the activation of transcription factor EB (TFEB), ultimately aiding in the parasite's survival (Fig. 6). Our findings add to the growing body of evidence illustrating how parasites reprogram host cells to their advantage.

## Methods

### Experimental animals at University of Bern
Female and male mice used in the experiments were between 6 and 12 weeks of age and were bred in the central animal facility of the University of Bern. We have complied with all relevant ethical regulations for animal use and we strictly followed the guidelines of the Swiss Tierschutzgesetz (TSchG; Animal Rights Laws). All experiments were approved by the ethical committee of the University of Bern (license number: BE86/19).

### Parasite strains
All parasite strains used have a *P. berghei* ANKA background. *Pb*mCherry, *Pb*GFP parasites are phenotypically wild type like *Pb*WT. *Pb*mCherry express cytosolic mCherry under the control of the *P. berghei* hsp70 regulatory sequences[43]. *Pb*GFP parasites express cytosolic GFP under the promoter of the eukaryotic elongation factor 1-alpha (eef1α)[44].

### Cell lines
Wild type HeLa cells (European Cell Culture Collection); wild type Huh7 cells (Japanese Collection of Research Bioresources Cell Bank JCRB0403); HeLa ATG16L1-KO, ATG16L1-KO + BFP-ATG16L1, ATG16L1-KO + BFP-ATG16L1-ΔWD40 were provided by Prof. Richard Youle; HeLa LC3-3KO (KO for LC3A, LC3B, LC3C), GAB-3KO (KO for GABARAP, GABARAPL1, GABARAPL2), LC3-GAB-6KO (KO for LC3A, LC3B, LC3C, GABARAP, GABARAPL1, GABARAPL2) were provided by Dr. Michael Lazarou[34]; HeLa FNIP1/2-DKO (KO for FNIP1 and FNIP2) were provided by Dr. Leon O. Murphy[21], FNIP1/2-DKO+3xHA-FNIP1, FNIP1/2-DKO+3xHA-FNIP1-LIR (both generated in Heussler lab); HeLa expressing SopF and all cell lines expressing TFEBmCh were generated for this study.

### Plasmids
GFP-GABARAP and GFP-GABARAPL1 constructs were made using the pEGFP-C1 plasmid from Clontech. GABARAP cDNA was amplified from pGex-4T-2_GABARAP, Addgene plasmid # 73948[45] using primers 5′-CTCGAGAAATGAAGTTCGTGTACAAAGAAGAG-3' and 5′-GGATCCTCACAGACCGTAGACACTTTC-3'. GABARAPL1 cDNA was amplified from pGex-4T-2_GEC1, Addgene plasmid #73945[46] using primers 5′-CTCGAGAAATGAAGTTCCAGTACAAGGAGG-3' and 5′-GGATCCTCATTTCCCATAGACACTCTCATC-3'. Both template plasmids were a gift from Dieter Willbold. Both cDNAs were subcloned, verified by sequencing and finally cloned into pEGFP-C1 using Xho1 and BamH1 restriction enzymes. The plasmid GFP-GABARAPL2 was a gift from Zvulun Elazar. pRK5-HA GST RagD77L was a gift from David Sabatini, Addgene plasmid # 19308[47]. FLCN-eGFP was a gift from David Sabatini, Addgene plasmid # 72289[48]. hFNIP1-WT and hFNIP1-Y583A/V586A both in TetLenti were a gift from Leon O. Murphy, Casma Therapeutics, Cambridge, MA, USA[21].

For generation of stable cell lines we used the PiggyBac plasmid system or a lentiviral system. For PiggyBac we mainly used the improved plasmid version pPB3.0-Blast which was a gift from Olivier Pertz, Bern, Switzerland. For stable DNA integration we transfected the PiggyBac plasmids together with the helper plasmid expressing a transposase[49]. The lentiviral expression plasmid pRRLSIN.cPPT. PGK-GFP.WPRE (Addgene plasmid #12252), as well as the VSV-G envelope-expressing plasmid pMD2.G (Addgene plasmid #12259) and the second-generation packaging plasmid psPAX2 (Addgene plasmid #12260) were gifts from Didier Trono.

## Culture, treatment and in vitro infection of HeLa and Huh7 cells and primary Hepatocytes

Wild type HeLa cells (European Cell Culture Collection) and Huh7 cells (Japanese Collection of Research Bioresources Cell Bank JCRB0403) were cultured in Minimum Essential Medium with Earle's salts (MEM EBS; BioConcept, 1-31F01-I), supplemented with 10% FCS (GE Healthcare), 100 U penicillin, 100 µg/ml streptomycin, and 2 mM L-glutamine (all from Bioconcept). Cells were cultured at 37 °C and 5% $CO_2$ and split using Accutase (Innovative Cell Technologies). For starvation, cells were rinsed 3 times with Earle's Balanced Salt Solution (EBSS, Gibco, 14155063) and subsequently incubated in this salt solution for 2 h before fixation.

Mouse primary hepatocytes were isolated as described elsewhere[7] and cultured in William's E medium (Bioconcept, 1-48F02-I) supplemented with 10% FCS (GE Healthcare), 100 U penicillin, 100 µg/ml streptomycin, and 2 mM L-glutamine (all from Bioconcept) at 37 °C and 5% $CO_2$. Torin 1 (Cell Signaling, 14379) treatment was carried out for 2 h at a concentration of 200 nM.

For infection of all used cells, salivary glands of *P. berghei*-infected *Anopheles stephensi* mosquitoes were isolated and disrupted to release sporozoites. Sporozoites were incubated with cells in the smallest possible volume of MEM EBS medium containing 25 µg/ml Amphotericin B (4-05F00-H BioConcept) for 1 h. Subsequently, they were rinsed and incubated in the respective medium.

## Transfection of HeLa cells

HeLa cells were harvested by Accutase treatment and $1 \times 10^6$ cells were pelleted by centrifugation at $1000 \times g$ for 5 min at room temperature. Cells were resuspended in Nucleofector V Solution (VVCA-1003, Lonza) and transfected with 1 µg of plasmid DNA using program T-028 of the Nucleofector 2b transfection device according to the manufacturer's instructions.

## Generation of stable cell lines

To generate HeLa cells constitutively expressing TFEB-mCherry, the TFEB-mCherry cDNA was amplified using primers 5′-ATCGGACCGATGG CGTCACGCATAG-3′ and 5′-ATCGGTCCGCTACTTGTACAGCTCG-3′ from a TFEB-mCherry expression vector produced in our Lab (pTFEB-mCherry, where TFEB cDNA was amplified using primers 5′-TTGGTAC CATGGCGTCACGCATAGGGTTG-3′ and 5′-AGAGGATCCGCCAGCA CATCGCCCTCCTCC-3′ using HeLa cDNA as template. The PCR product was subcloned, sequenced and then finally cloned into pmCh-N1 via Kpn1 and BamH1 restriction sites), and subcloned via RsrII restriction sites in pRRLSIN. cPPT.PGK-GFP.WPRE. Virus production and transduction of cells were done as described previously for GFP-LC3B-expressing cells[50]. Fluorescent-based cell sorting was used to enrich TFEBmCherry expressing cells.

For generation of HeLa cell lines stably expressing 3x-HA-hFNIP1 and 3x-HA-FNIP1-LIR we used the PiggyBac plasmid system. 3x-HA-FNIP1 and 3xHA-FNIP1-LIR were amplified with primers 5′-CGCTCGA-GATGTATCCCTATGACGTG-3′ and 5′-CGCTCGAGTTAAAGGAG-TATTTGTGCAAC-3′ using hFNIP1-WT and hFNIP1-Y583A/V586A (FNIP1-LIR) both in TetLenti as templates. Both cDNAs were subcloned into pJet1.2/blunt, verified by sequencing and then finally cloned into pPiggyBac3.0-Blast using Xho1 restriction sites. Transfected cells were selected using 4 µg/ml Blasticidin (InvivoGen, ant-bl-05) for 48 h.

To generate HeLa cells constitutively expressing SopF-V5, the SopF open reading frame was amplified using primers 5′-GCTAGCATGCT CAAACCTATCTGCC-3′ and 5′-CCCGGGATATAATATTATGCAGTC TCTATT-3′ using pmCherry-SopF as template. The cDNA was subcloned into pJet1.2/blunt. After sequencing, the SopF fragment was cloned into the pLX307 lentiviral vector using restriction sites NheI and SmaI. Virus production and transduction of cells were done as described previously for GFP-LC3B-expressing cells[50]. Cells were selected from 2 days after lentiviral transduction with 1 µg/ml Puromycin for 3 days. Single-cell clones were obtained and confirmed by IFA using an anti-V5 antibody.

## Protein lysates and western blotting

Cells were grown in MEM medium in a 25 $cm^2$ flask, detached with Accutase and 1 million cells were spun down and washed 1 times with PBS. The cell pellet was resuspended in 80 µl water. To lyse the cells 20 µl 5x concentrated Laemmly sample buffer was added (final concentration 1x Laemmli buffer, 2% glycerol, 25 mM Tris HCl pH 6.8, 0.8% SDS, 0.004% bromophenolblue, 2% 2-mercaptoethanol). The lysate was then treated with universal nuclease (Thermo Fisher Scientific, 88701) for 5 min at room temperature. Proteins were denatured at 95 °C for 5 min. To detect TFEB, the proteins were separated on 12% SDS PAGE. PageRuler Prestained NIR Protein Ladder (Thermo Fisher Scientific, 26635) was used as a molecular weight marker. The transfer to nitrocellulose membranes was performed in a tank blot device (Hoefer). Five percent fat-free milk in TBS (Tris-buffered saline; 10 mM Tris, 150 mM NaCl) was used for blocking the membrane. Antibodies were diluted in 5% milk/TBST. Antibodies used were rabbit anti-TFEB (Cell Signaling Technology, #4240, 1:1000), and mouse anti-α-tubulin (Sigma-Aldrich, T6199, 1:1000). For secondary antibody incubation, anti-rabbit IgG 800 CW IRDye and anti-mouse IgG 680 LT IRDye (Li-Cor Biosciences, all 1:10,000) were used. A Li-Cor Odyssey Imaging system (Li-Cor Biosciences) was used for detection.

## Indirect immunofluorescence analysis (IFA)

Cells were grown on glass cover slips and after indicated time periods fixed with 4% paraformaldehyde in phosphate-buffered saline (PBS; 137 mM NaCl, 2.7 mM KCl, 10 mM $Na_2HPO_4$, 1.8 mM $KH_2PO_4$, pH 7.4) for 10 min (all incubations at room temperature). Permeabilization was performed for 5 min in 0.1% Triton X-100 (T8787, Fluka-Chemie) in PBS. After washing with PBS, unspecific binding sites were blocked by incubation in 10% FCS/PBS or in 1% BSA/PBS for 10 min followed by incubation with primary antibody in 10% FCS-PBS or in 1% BSA/PBS for at least 1 h. After washing with PBS, cells were incubated with fluorescently labeled secondary antibodies diluted 1:1000 in 10% FCS/PBS for 1 h. DNA was stained with Dapi solution 1 µg/ml in PBS for 5 min. Cover slips were mounted onto glass slides using ProLong™ Gold antifade solution (P36930, Invitrogen).

Primary antibodies used were, rat monoclonal anti-red antibody (Chromotek 5f8; 1:2000), rabbit polyclonal anti-GFP antibody (Origene, SP3005P; 1:1000), rabbit monoclonal anti-GFP antibody (Cell Signaling, CS2956; 1:1000), mouse monoclonal anti-GFP antibody (Roche 11814460001; 1:1000), rabbit monoclonal anti-pS6Ser240/244 antibody (Cell Signaling, CS5364; 1:1000), rabbit polyclonal anti-pS6Ser235/236 antibody (Cell Signaling, CS2211; 1:1000), rabbit monoclonal anti-p4E-BP1Thr37/46 antibody (Cell Signaling, CS2855; 1:1000), mouse monoclonal anti-V5 antibody (Invitrogen R960-25, 1:1000), rabbit anti-UIS4 antiserum (1:5000), anti-GABL1 ankyron protein (Proimmune, Ankyron1191, AH50324; 1:500), mouse monoclonal anti-HA antibody (Santa Cruz, SC-7392, 1:500). Secondary antibodies used were anti-rat Alexa594 (Invitrogen A-11007; 1:1000), anti-rabbit Alexa488 (Invitrogen A-11008; 1:1000), anti-mouse Alexa488 (Invitrogen A-11001), anti-rabbit Cy5 (Dianova; 1:1000), anti-mouse Cy5 (Dianova; 1:1000).

## Microscopy and quantifications

Confocal images were acquired with an inverted Leica TCS SP8 using an HC PL APO CS2 63×/1.4 NA immersion oil objective. Pearson's correlation coefficient (PCC) was calculated using the Coloc2 tool of the FIJI software. Image processing was performed using FIJI. Quantification of GFP-GABARAP-positive or 3xHA-FNIP1-positive parasites was done by visual inspection using a Leica DM5500B epifluorescence microscope. Only UIS4-positive parasites were considered. Parasites were counted positive if a clear association between the protein of interest and UIS4 at the parasite circumference was observed which means that more than 30% of the proximity of the parasite was covered with the protein of interest.

For quantification of TFEB localization, pictures were taken using the Leica DM5500B epifluorescence microscope. The fluorescence intensity of the TFEBmCh signal in the cytosol and nucleus was measured using FIJI. An area of $4.13 \times 4.13$ µm² was measured in both compartments whereas the

cytosolic measurement was taken as close to the nucleus as possible because the cytosolic signal quickly decreased toward the periphery in most cells. The ratio of the nuclear over the cytosolic fluorescence intensity signal of the same cell was calculated using Microsoft Excel.

For quantification of mTORC1 activity, pictures were taken with a Leica DM5500B epifluorescence microscope using a 63x HCX PL APO objective. The fluorescence intensity of the cytosolic pS6 and p4E-BP1 signals was measured using FIJI. An area of $4.13 \times 4.13~\mu m^2$ was measured.

Automated live cell imaging was used to determine parasite size and numbers. Cells were seeded in a 96-well format and infected with mCherry- or GFP-expressing parasites. The same cells were imaged with an INCell Analyzer 2000 automated live cell imaging system (GE Healthcare Life Sciences) at 6, 24, and 48 hpi with minimal light exposure (15–25 ms). INCell Developer Toolbox 1.10.0 software was used to analyze the acquired images. The mCherry or GFP cytoplasmic signal was used to determine the parasite area. Segmentation was done using the "object" mode in the mCherry channel, and post-processing 2 was done to exclude objects smaller than $10~\mu m^2$.

## Statistics and reproducibility

All experiments were repeated at least three times independently unless otherwise indicated in the legend of the respective figure. The total number of experiments and sample size analyzed are indicated in the respective figure legend. Mean and standard deviations are indicated. All experimental conditions were statistically analyzed by either a two-tailed, unpaired Student's t test when comparing two groups only or a one-way ANOVA test when comparing multiple conditions. ANOVA tests were coupled to Tukey's post hoc test to analyze pairwise conditions or to Dunnett's test to compare different groups to the same control. GraphPad Prism version 9 was used to perform statistical analysis and draw graphs.

## Reporting summary

Further information on research design is available in the Nature Portfolio Reporting Summary linked to this article.

## Data availability

All data that support the findings of this study are available in the article and in the Supplementary Information. An uncropped picture of the Western Blot shown in Fig. 5 is shown in the Supplementary Information. Numerical source data for graphs can be found in Supplementary Data 1. All other data supporting this study's findings are available on request from the corresponding author.

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

## Acknowledgements

We thank Prof. Dr. Richard Youle for providing the HeLa ATG16L1-KO, ATG16L1-KO + BFP-ATG16L1, ATG16L1-KO + BFP-ATG16L1-ΔWD40 cell lines. Dr. Michaela Bulloch is thanked for critically reading the manuscript. We thank the MIC (Microscopy Imaging Centre) in Bern for providing excellent imaging facilities and technical support. This work was funded by the Swiss National Science Foundation (SNSF) (grant number 310030_182465) to Volker Heussler. Rebecca Cooper Foundation Fellowship, RC20241396, to Michael Lazarou.

## Author contributions

J.S., A.F.B., O.M.W., R.W., and R.R. conducted the experiments. T.L. and L.O.M. provided the HeLa FNIP1/2-DKO cell line and plasmids containing the FNIP1WT and the FNIP1-LIR cDNAs. M.L. and T.N.N. generated the HeLa LC3-3KO, GAB-3KO, and LC3-GAB-6KO cell lines. The HeLa TFEB-KO cell line was generated by J.M. and A.B. V.H. supervised and coordinated the project. J.S., A.F.B., R.W., O.W., and V.H. analyzed the data. J.S., A.F.B., and V.H. conceived and designed experiments. J.S., A.F.B., O.W., and V.H. wrote the paper.

## Competing interests

The authors declare the following competing interests: L.O.M. and T.L. are employees and shareholders of Casma Therapeutics. A.B. is a co-founder and shareholder of CASMA Therapeutics, Inc, and Advisory board member of Avilar Therapeutics and Amplify Therapeutics. L.O.M. is an author on a patent related to TFEB modulation, in addition. M.L. is a co-founder and member of the scientific advisory board of Automera. Other authors declare no competing interests.
