## [Transparent Peer Review file · Communications Biology]

Plasmodium berghei liver stage parasites exploit host GABARAP proteins for TFEB activation.

Corresponding Author: Professor Volker Heussler

Version 0:

Reviewer comments:

Reviewer #1

(Remarks to the Author)

In this interesting study the authors show activation of the transcription factor TFEB in response to Plasmodium liver stage infection. They further show that this activation follows a ATG16L1- and GABARAP-dependent pathway that has been described previously to occur in response to different forms of autophagy. They demonstrate that this described TFEB activation is occurring through sequestration of FLCN-FNIP at the parasite PVM in a GABARAP dependent manner. This study includes novel findings for the Plasmodium field, as it was not previously known that this transcription factor influencing lysosomal flux was activated during infection. That this pathway of TFEB activation is taken advantage of by Plasmodium parasites will be of interest to others interested in the link between autophagy and lysosomal regulation.

In general I think the conclusions are well supported.

Major comments:

When the greatest effect on TFEB localization comes at 24h pi, why was the localization of the GABARAPs and FNIP assessed so early in infection? At 6 hpi, the effect on TFEB localization in most infected cells is pretty minimal. Are these proteins present on parasite vacuoles at the time of the greatest effects on TFEB?

Page 8: "These findings suggest that mTORC1 activity towards TFEB is inhibited within the setting of P. berghei infection, thus leading to TFEB nuclear translocation, whereas mTORC1 activity towards canonical substrates S6 and 4EBP1 remained unchanged." I'm a little confused by this conclusion. It was stated in the introduction that TFEB activation can occur in a manner not reliant on mTORC1 signaling, and the authors have not shown an inhibition of mTORC1 activity.

Page 9: "...operates through a distinct, noncanonical mechanism that is dependent on a non-canonical mechanism which depends specifically on GABARAP ATG8 proteins and the CASM pathway...." Probably not exactly how the authors intended to phrase this. But the data seems to align well with what has been described for the other examples that rely on the CASM pathway – so how distinct is it then? I recommend rephrasing.

Page 20: "We unveil a novel molecular mechanism" My concern with this is similar to what is indicated above. It seems the parasites are taking advantage of a pathway that the cells already have and that has already been described to occur in several cases including CASM, xenophagy and mitophagy. I suggest rephrasing (and also keeping in mind that "non-canonical" just means discovered later).

Page 13: "...cells expressing FNIP1 with the mutated LIR motif... never show nuclear TFEB localization, even in P. berghei-infected cells" This statement is not quite true. The P value is even <0.005 when comparing infected and uninfected FNIP-LIR-expressing cells. I just recommend rephrasing.

Figure 6 seems to be not quite right and may be confusing. The scale of the parasite is an odd choice, but okay if the authors prefer that. But I thought the idea is that FLCN/FNIP recruited to the PVM are not available to act on RagC/D, and I think this is not really shown here.

Minor comments

The addition of line numbers would greatly facilitate review.

Some labeling on figure 1D is missing

Better attention could be paid to references. Only for example: page 7, start of paragraph 2.

I found it confusing that figure S3 was split up over multiple pages. Why not have an additional supplementary figure?

Page 11: "Under normal conditions, the FNIP-FLCN complex localizes at the lysosome" It was my understanding that FNIP-FLCN was recruited to the lysosome specifically by e.g. lysosomal damage / TRPML1 activation.

Page 12: "...and underscores the utility of FNIP1/2-DKO cell lines..." It seems like these localization results (of tagged FNIP1) would be the same with or without endogenous FNIP1/2 - so I don't entirely understand this phrase. (Perhaps it is just in the wrong location.)

Reviewer #2

(Remarks to the Author)

Plasmodium parasites encounter numerous challenges during their life cycle, particularly during the liver stage, where they proliferate rapidly. These challenges include acquiring sufficient nutrients, evading the host cell's intracellular immune response, and inhibiting host cell death to ensure parasite survival. Plasmodium parasites overcome these obstacles by actively manipulating host cell processes. Present study explored the activation of host cell TFEB in Plasmodium berghei-infected cells during the liverstage of the parasite. Results show a critical role of proteins belonging to the GABARAP subfamily of ATG8 proteins in recruiting the TFEB-blocking FLCN-FNIP complex to the PVM. Authors demonstrated that the sequestration of FLCN-FNIP resulted in a robust activation of TFEB, independent of the nutritional status of the cell but reliant on CASM and GABARAP proteins. Findings in the work presented provide novel mechanistic insights into host cell signalling occurring at the PVM, during the host parasite interactions. This work provides an example of how P. berghei manipulates host cell signalling at its PVM.

Overall, the work is thorough, well planned and well executed. Data support the conclusions without any doubts. The statistics applied is suitable for the conclusions drawn. The work is of general interest for parasitologists, and people interested in host parasite interactions. The reviewer complements the authors for excellent work as well as for equally good presentation of the data.

Specific comments:

1- Figure 1 and 3-legend: "ration" should be corrected to "ratio"

Reviewer #3

(Remarks to the Author)

Malaria, caused by the Plasmodium parasite, is the deadliest parasitic disease worldwide, resulting in over 600,000 deaths annually. Although a vaccine is available, its limited efficacy and the emerging resistance to treatments and insecticides necessitate ongoing research to better understand the parasite and develop new treatments. In this context, Schmuckli-Maurer et al. investigated the activation of the host cell TFEB in Plasmodium berghei-infected cells during the liver stage of the parasite, in their article entitled "Parasitic paradox: Plasmodium berghei activates host anabolic and catabolic pathways".

Their results revealed that GABARAP proteins, integrated into parasitophorous vacuolar membrane (PVM) through a process involving CASM, play a key role in recruiting the TFEB-blocking FLCN-FNIP complex. This recruitment leads to strong activation of TFEB, independent of the cell's nutritional status but dependent on CASM and GABARAP proteins. This mechanism appears broadly similar to previous findings in Salmonella infections or upon disruption of the lysosome, which diminishes the novelty of the article. One of the key take-home messages is that in cells infected by P. berghei, both anabolic and catabolic processes are simultaneously activated.

Overall, this article is enjoyable to read and flows smoothly. The reader appreciates the clarity of the explanations, whether in the introduction or when presenting the results. The presence of the model is highly valuable, although it could be better utilized (Figure 6 is never mentioned in the text).

One of my major concerns is the absence of verification of the effect of knockdown (KD) and subsequent re-complementation. It is crucial, if not essential, to assess the expression levels of these proteins (ATG16L1, LC3, GABARAP, etc.) both before and after re-complementation, using Western Blot and/or Immunofluorescence.

This article could also be improved by addressing the following minor concerns :

- The chosen title effectively summarizes one of the key messages of this article, but it may not fully represent the concepts

discussed, potentially hindering the good referencing and thus, the effective dissemination of the article (TFEB, GABARAP, ATG8ylation, CASM, etc.).

- In the abstract, you use "parasitophorous vacuole membrane" while you only use "parasitophorous vacuole" in the main text. Please ensure coherence in your terminology throughout.

- Overall, there are a few concepts used throughout the text (such as autophagy, selective autophagy, canonical or non-canonical) without proper introduction. I understand this is for the sake of simplicity, but it can be lacking.

- In the introduction:

o The reference number 6 appears somewhat specialized for the associated message. Could you provide justification for this choice or consider replacing it with a reference from a more general review?

o The first occurrence (not the following ones) of reference 11 is misleading because it concerns Salmonella and not Plasmodium, both of which are pathogens that generate a vacuole. If this is not a referencing issue, please clarify the context.

o Could you please verify the accuracy of reference 15? It appears that, even though it discusses the lysosome, it does not mention TFEB.

- For the figures/results:

o It would be easier to interpret if all panels were annotated with the proteins stained (e.g., Fig. 1A: Could the authors add TFEB in red and the parasites in green, even though this information is already noted in the legend and text?). This applies to Fig. 1, 2, and S2.

o In Fig. 1B, the panel showing the control of cells grown in full medium is missing.

o In Fig. 1C, could the authors please complete the x-axis label by adding 'Pb infected' as in the graphs of Figure S1?

o Overall, it would be important to clarify in the quantification whether the authors specifically count infected cells or they choose random fields for quantification. Additionally, the low number of infected cells in the microscopy images raises questions. It may be worthwhile to explain this situation (e.g., technical limitations, multiplicity of infection, etc.) for people who are not from the field.

o In Fig. 2B and its quantification, please consider organizing the panels in the order they are mentioned in the text for more clarity: ATG16L1 KO ATG-16L1-DeltaWD40 ATG16L1-B.

o In Fig. 3B, in my opinion, the number of replicates is a bit low, especially considering that N>30 for every other experiment in the article.

o It would be interesting for the author to justify the choice of 6 hours post-infection (hpi) for conducting the experiments.

Version 1:

Reviewer comments:

Reviewer #1

(Remarks to the Author)

The authors addressed my concerns. The additional experiments, clarifications and re-phrasing of some key points improved this already engaging study.

Reviewer #3

(Remarks to the Author)

The authors have carefully considered the reviewers' comments and have responded point by point. Each suggestion has been thoroughly addressed and the recommended changes have been incorporated into the manuscript, significantly strengthening the scientific rigor and validity of the study.

Point by point response to the reviewers

We sincerely thank the reviewers for their time and expertise in thoroughly evaluating our manuscript. Their insightful comments and valuable feedback have been instrumental in enhancing the quality and clarity of our work. We have carefully considered each suggestion and incorporated the recommended changes into the manuscript. We believe that addressing these suggestions has significantly strengthened the scientific rigor and validity of our study.

Reviewer #1 (Remarks to the Author):

In this interesting study the authors show activation of the transcription factor TFEB in response to *Plasmodium* liver stage infection. They further show that this activation follows a ATG16L1- and GABARAP-dependent pathway that has been described previously to occur in response to different forms of autophagy. They demonstrate that this described TFEB activation is occurring through sequestration of FLCN-FNIP at the parasite PVM in a GABARAP dependent manner. This study includes novel findings for the *Plasmodium* field, as it was not previously known that this transcription factor influencing lysosomal flux was activated during infection. That this pathway of TFEB activation is taken advantage of by *Plasmodium* parasites will be of interest to others interested in the link between autophagy and lysosomal regulation.

In general I think the conclusions are well supported.

Major comments:

When the greatest effect on TFEB localization comes at 24 hpi, why was the localization of the GABARAPs and FNIP assessed so early in infection? At 6 hpi, the effect on TFEB localization in most infected cells is pretty minimal. Are these proteins present on parasite vacuoles at the time of the greatest effects on TFEB?

Thank you for this important and valid question. We would like to clarify our reasoning for assessing the localization of GABARAPs and FNIP at 6 hpi:

- As shown in Fig. 1c, the nuclear translocation of TFEB in *Plasmodium*-infected cells is already significantly evident at 6 hpi.
- While TFEB nuclear localization is indeed most prominent at 24 hpi, this may be attributed to the increased parasite size and larger surface area at this later stage, allowing more FNIP1 to accumulate at the PVM. However, the underlying molecular mechanisms remain consistent, as the proteins involved are present at both 6 hpi and 24 hpi.
- In some experiments, we used transiently transfected cells. Given the low infection rate with *P. berghei* (between 1% and 5%), it is particularly challenging to obtain cells that are both transfected and infected simultaneously. We found it more feasible to achieve a reasonable number of infected and transfected cells at 6 hpi. Given these challenges and for consistency, we chose to study the molecular mechanisms at 6 hpi and to conduct our microscopy quantification at this timepoint, rather than performing some experiments at 6 hpi and others at 24 hpi.
- To address the reviewer's concerns, we have conducted additional experiments at 24 hpi and have included these results in the revised manuscript. Specifically, we have added IFAs demonstrating that GABARAPs localize at the parasite PVM 24 hpi. **Figure S4c** shows that endogenous GABARAPs (stained with the GABARAP-specific Ankyron) localize at the

parasite's PVM at 24 hpi. Additionally, as shown in **Figure S5**, ectopically expressed GFP-GABARAPs associate also associate with the PVM at 24 hpi. These findings further support our conclusions regarding the role of GABARAPs in the context of *Plasmodium* infection.

Figure S4 Endogenous GABARAPs localis to the PVM

Figure S5 Expression of GFP-GABARAPs in GAB-3KO cells

Page 8: “These findings suggest that mTORC1 activity towards TFEB is inhibited within the setting of *P. berghei* infection, thus leading to TFEB nuclear translocation, whereas mTORC1 activity towards canonical substrates S6 and 4EBP1 remained unchanged.” I’m a little confused by this conclusion. It was stated in the introduction that TFEB activation can occur in a manner not reliant on mTORC1 signaling, and the authors have not shown an inhibition of mTORC1 activity.

Line 164

Thank you for pointing out this confusion. We agree that this observation may seem puzzling at first, and it wasn’t fully clear to us until we were able to resolve the underlying molecular mechanism. mTORC1 plays a dual role in controlling cell metabolism: it promotes anabolic processes such as protein, nucleotide and lipid synthesis by phosphorylating substrates like S6K and transcription factor 4E-BP1, while simultaneously inhibiting catabolic processes, including autophagy and lysosomal biogenesis, through phosphorylation of TFEB.

- For a long time, it was assumed that mTORC1 phosphorylates all its substrates uniformly. However, recent studies have demonstrated that different upstream stimuli can result in selective downstream responses, a phenomenon called non-canonical mTORC1 signaling.
- This selective mTORC1 regulation is precisely what we observe in our experiments. In *P. berghei*-infected cells, mTORC1 is not inhibited overall; it remains active on substrates that support cell anabolism, such as S6K and 4E-BP1. However, it selectively fails to phosphorylate TFEB, which would normally inhibit catabolic processes.

- Therefore, our findings indicate that mTORC1 signaling in infected cells is selectively altered, consistent with non-canonical mTORC1 regulation.

To clarify this in the manuscript, we have revised the text to better explain how mTORC1 activity is differentially regulated in the context of *P. berghei* infection, particularly in relation to TFEB phosphorylation. We hope this explanation clarifies our conclusions and the basis for our observations.

Page 9: "...operates through a distinct, noncanonical mechanism that is dependent on a non-canonical mechanism which depends specifically on GABARAP ATG8 proteins and the CASM pathway...." Probably not exactly how the authors intended to phrase this. But the data seems to align well with what has been described for the other examples that rely on the CASM pathway – so how distinct is it then? I recommend rephrasing.

Line 234

- Thank you for pointing out the ambiguity in our original phrasing. We agree that the sentence was unclear and potentially confusing. To address this, we have revised the sentence for clarity. The revised sentence now reads: This suggests that TFEB activation in *P. berghei*-infected cells is mediated by a substrate-specific, non-canonical mTORC1 signaling pathway that specifically depends on GABARAP ATG8 proteins and the CASM pathway.

We believe this rephrasing more accurately reflects the nature of the signaling pathway involved and aligns with existing descriptions of the CASM pathway.

Page 20: "We unveil a novel molecular mechanism" My concern with this is similar to what is indicated above. It seems the parasites are taking advantage of a pathway that the cells already have and that has already been described to occur in several cases including CASM, xenophagy and mitophagy. I suggest rephrasing (and also keeping in mind that "non-canonical" just means discovered later).

Line 613

Thank you for your observation. We acknowledge that the molecular mechanism involving FLCN/FNIP and GABARAPs has been previously described by others.

- In our study, we present a new context in *P. berghei*-infected cells, demonstrating how the parasite recruits the FLCN/FNIP complex to its surface and thereby interferes with substrate-specific non-canonical mTORC1 signaling.
- To better reflect this, we have rephrased the sentence as follows: We describe the molecular mechanism by which the parasite hijacks the FNIP-FLCN complex via the host cell's GABARAP proteins, which are incorporated into the PVM

We believe this revised wording more accurately conveys our contribution while acknowledging the existing knowledge in the field.

Page 13: "...cells expressing FNIP1 with the mutated LIR motif... never show nuclear TFEB localization, even in *P. berghei*-infected cells" This statement is not quite true. The P value is even <0.005 when comparing infected and uninfected FNIP-LIR-expressing cells. I just recommend rephrasing.

Line 386

Thank you for highlighting this point. We agree that the original statement was not entirely accurate.

- We have revised the sentence to better reflect the data: In contrast, cells expressing FNIP1 with a mutated LIR motif (3xHA-FNIP1-LIR), which cannot interact with GABARAPs and thus cannot be sequestered at the PVM, rarely exhibit more nuclear than cytoplasmic TFEB, even in *P. berghei*-infected cells (Fig. 4d, S7a).

We believe this rephrasing more accurately represents our findings and addresses your concern.

Figure 6 seems to be not quite right and may be confusing. The scale of the parasite is an odd choice, but okay if the authors prefer that. But I thought the idea is that FLCN/FNIP recruited to the PVM are not available to act on RagC/D, and I think this is not really shown here.

Thank you for your feedback on Figure 6. We agree that the original figure may have been unclear.

- We have revised Figure 6 to enhance clarity. The parasite is now depicted at a larger scale, and to clearly indicate that FLCN/FNIP can no longer activate RagC/D, we have marked the activation arrow with a red cross.

We hope these changes improve the figure's accuracy and make the concept more understandable.

Figure 6 *Plasmodium berghei* manipulates mTORC1 signaling

Minor comments

The addition of line numbers would greatly facilitate review.

We completely agree, and we apologize for the oversight. We have now added line numbers to the manuscript to facilitate easier review.

Some labeling on figure 1D is missing

- We have now added the missing labeling in Figure 1d.

Figure 1 TFEB translocates to host cell nucleus upon *P. berghei* infection

Better attention could be paid to references. Only for example: page 7, start of paragraph 2.

Line 155

- Thank you for this comment. We have now added the reference of Napolitano et al., (2022) at the suggested place in the manuscript. Napolitano, G., Malta, C. D. & Ballabio, A. Non-canonical mTORC1 signaling at the lysosome. *Trends Cell Biol* (2022) doi:10.1016/j.tcb.2022.04.012.

I found it confusing that figure S3 was split up over multiple pages. Why not have an additional supplementary figure?

- We apologize for the confusion caused by the split of Figure S3. We have addressed this issue by reorganizing the figures: Figure S3c, d, e is now presented as new Figure S4a, b, d. The new Figure S4 shows in addition GABARAP proteins which localise to the parasite PVM 24 hpi

(panel c). The former Figure S4 has been renumbered as Figure S6. The corresponding changes have been made in the text as well.

Figure S3 PVM localisation of GABARAPs is inhibited by SopF

Figure S4 Endogenous GABARAPs localise to the PVM

Page 11: “Under normal conditions, the FNIP-FLCN complex localizes at the lysosome” It was my understanding that FNIP-FLCN was recruited to the lysosome specifically by e.g. lysosomal damage / TRPML1 activation.

Line 332

- We have rephrased the sentence in the manuscript to more accurately reflect the function of the FNIP-FLCN complex: Under typical conditions, the FNIP-FLCN complex functions as a GAP for RagC/D with GDP-bound RagC/D promoting the interaction with TFEB at the lysosome surface.

Page 12: “...and underscores the utility of FNIP1/2-DKO cell lines...” It seems like these localization results (of tagged FNIP1) would be the same with or without endogenous FNIP1/2 - so I don't entirely understand this phrase. (Perhaps it is just in the wrong location.)

Line 369

Thank you for your comment. You are correct that the localization of tagged FNIP1 would be similar regardless of the presence of endogenous FNIP1/2. However, the results are clearer when only one version of FNIP1 is expressed in the cell, as it eliminates competition between endogenous FNIP1 and HA-tagged FNIP1.

- We agree that the sentence was misplaced in the original text, and we removed it from this paragraph. Specifically, for the results shown in Fig. S6 regarding the recruitment of FLCN to the parasite's PVM, it is crucial to express HA-FNIP1-LIR without any competing endogenously expressed wild-type FNIP1.

- We still believe that the FNIP1/2-DKO cell line is very useful for our experiments, particularly for studying the FNIP-LIR mutant. Therefore, we have added the sentence to the end of the next paragraph, which describes the localization of FLCN: These findings further validate the utility of FNIP1/2-DKO cell lines in dissecting the molecular interactions within the FNIP-FLCN complex at the PVM.

We hope this adjustment clarifies our intent and appropriately positions the statement within the manuscript.

Reviewer #2 (Remarks to the Author):

Plasmodium parasites encounter numerous challenges during their life cycle, particularly during the liver stage, where they proliferate rapidly. These challenges include acquiring sufficient nutrients, evading the host cell's intracellular immune response, and inhibiting host cell death to ensure parasite survival. Plasmodium parasites overcome these obstacles by actively manipulating host cell processes. Present study explored the activation of host cell TFEB in Plasmodium berghei-infected cells during the liverstage of the parasite. Results show a critical role of proteins belonging to the GABARAP subfamily of ATG8 proteins in recruiting the TFEB-blocking FLCN-FNIP complex to the PVM. Authors demonstrated that the sequestration of FLCN-FNIP resulted in a robust activation of TFEB, independent of the nutritional status of the cell but reliant on CASM and GABARAP proteins. Findings in the work presented provide novel mechanistic insights into host cell signalling occurring at the PVM, during the host parasite interactions. This work provides an example of how P. berghei manipulates host cell signalling at its PVM.

Overall, the work is thorough, well planned and well executed. Data support the conclusions without any doubts. The statistics applied is suitable for the conclusions drawn. The work is of general interest for parasitologists, and people interested in host parasite interactions. The reviewer complements the authors for excellent work as well as for equally good presentation of the data.

We sincerely thank the reviewer for their thoughtful and encouraging comments. We are pleased that the reviewer found our study to be thorough, well-executed, and of broad interest to the scientific community. We appreciate the recognition of our efforts to present the data clearly and effectively. Your positive feedback motivates us to continue our research in host-parasite interactions with the same rigor and enthusiasm.

Thank you again for your valuable review and support of our work.

Specific comments:

1- Figure 1 and 3-legend: "ration" should be corrected to "ratio"

- Thank you for pointing out this error. We have corrected "ration" to "ratio" in the legends of Figures 1 and 3.

Reviewer #3 (Remarks to the Author):

Malaria, caused by the Plasmodium parasite, is the deadliest parasitic disease worldwide, resulting in over 600,000 deaths annually. Although a vaccine is available, its limited efficacy and the emerging resistance to treatments and insecticides necessitate ongoing research to better understand the parasite and develop new treatments. In this context, Schmuckli-Maurer et al. investigated the activation of the host cell TFEB in Plasmodium berghei-infected cells during the liver stage of the parasite, in their

article entitled “Parasitic paradox: *Plasmodium berghei* activates host anabolic and catabolic pathways”.

Their results revealed that GABARAP proteins, integrated into parasitophorous vacuolar membrane (PVM) through a process involving CASM, play a key role in recruiting the TFEB-blocking FLCN-FNIP complex. This recruitment leads to strong activation of TFEB, independent of the cell's nutritional status but dependent on CASM and GABARAP proteins. This mechanism appears broadly similar to previous findings in *Salmonella* infections or upon disruption of the lysosome, which diminishes the novelty of the article. One of the key take-home messages is that in cells infected by *P. berghei*, both anabolic and catabolic processes are simultaneously activated.

Overall, this article is enjoyable to read and flows smoothly. The reader appreciates the clarity of the explanations, whether in the introduction or when presenting the results. The presence of the model is highly valuable, although it could be better utilized (Figure 6 is never mentioned in the text).

One of my major concerns is the absence of verification of the effect of knockdown (KD) and subsequent re-complementation. It is crucial, if not essential, to assess the expression levels of these proteins (ATG16L1, LC3, GABARAP, etc.) both before and after re-complementation, using Western Blot and/or Immunofluorescence.

Thank you for raising this important point. We understand the concern regarding the verification of protein expression levels in the various cell lines used in our study.

- All cell lines we utilized are knock-out cell lines, and not knockdown lines.
- The HeLa ATG16L1 **knock out** cells, as well as ATG16L1^{-/-} cells constitutively expressing ATG16L1-β-BFP or ATG16L1-ΔWD40-BFP, were originally generated and characterized in Fischer et al., 2020. These cell lines were further characterized in our lab after receiving them from R. Youle’s lab, and our findings were published in Bindschedler et al., 2023, specifically in Fig. 2h (see picture).

- The HeLa cell lines knock-out for either all three LC3s (LC3A, LC3B, LC3C), all three GABARAPs (GABARAP, GABARAPL1, GABARAPL2) or all six ATG8s (LC3A, LC3B, LC3C, GABARAP, GABARAPL1, GABARAPL2) were generated in Michael Lazarou’s lab and published in Nguyen et al., 2015.
- The TFEBmCherry-expressing cell lines were generated in our lab and TFEBmCherry expression levels are visible in the IFA images. Although TFEBmCherry is not expressed at uniform levels across all cells, this variation doesn’t impact our results, as the TFEB quantification was based on the ratio of nuclear versus cytoplasmic TFEB.

- We have now added a Western Blot in Figure 5c. It shows the TFEB protein levels in HeLa WT, HeLa TFEB knock-out cells and TFEB knock-out cells expressing TFEBmCherry. These cell lines were used to study parasite fitness (Data shown in Figure 5a and 5b).

Figure 5 Parasite Fitness in TFEB-KO cells

- In addition, we have included new IFAs of GAB-3KO cells transiently transfected with each of the GABARAPs as GFP fusion proteins, which were used in the quantification presented in Figure 3e. These results are now presented in the new Figure S5. This figure shows the expression levels of the different GFP-GABs and demonstrates that all three GABARAPs localize at the parasite's PVM 24 hours post infection.

Figure S5 Expression of GFP-GABARAPs in GAB-3KO cells

- To further strengthen our findings and provide additional clarity, we have included a new Figure S7 with the following details:

Figure S7a: This panel shows the expression levels of 3xHA-FNIP and 3xHA-FNIP-LIR in stably transfected HeLa FNIP1/2-DKO cells, as observed in immunofluorescence assay (IFA) images. These cell lines were used for the quantifications presented in Figures 4d and 4e.

Figure S7b: This panel displays the expression levels of transiently transfected RagD77L cells, which were also used in the experiments shown in Figures 4d and 4e.

Figure S7c: This panel shows TFEBmCherry expression in non-treated and EBSS-treated HeLa FNIP1/2-DKO cell lines used in the experiment shown in Figure 4e.

These additional data provide a clearer picture of the expression levels in the relevant cell lines, further validating the results presented in the manuscript.

Figure S7 Expression of 3xHA-FNIP1, 3xHA-FNIP1-LIR and HA-RagD77L in FNIP1/2-DKO cells

This article could also be improved by addressing the following minor concerns :

- The chosen title effectively summarizes one of the key messages of this article, but it may not fully represent the concepts discussed, potentially hindering the good referencing and thus, the effective dissemination of the article (TFEB, GABARAP, ATG8ylation, CASM, etc.).

- Thank you for your thoughtful feedback regarding the title. We understand your concern about the potential impact on referencing and dissemination. However, we believe that the title is both engaging and memorable, which can attract a broader audience. Additionally, since the key terms (TFEB, GABARAP, ATG8ylation, CASM, etc.) are prominently featured in the abstract, we are confident that readers will have a clear understanding of the specific concepts discussed in the article.

We appreciate your suggestion and believe that the combination of a catchy title and a detailed abstract will effectively communicate the content of our work.

- In the abstract, you use “parasitophorous vacuole membrane” while you only use “parasitophorous vacuole” in the main text. Please ensure coherence in your terminology throughout.

Line 30

- Thank you for pointing out this inconsistency. We have corrected the terminology in the abstract to ensure coherence with the main text. The revised sentence now reads: **During this liver stage development, parasites reside in a parasitophorous vacuole (PV) whose membrane acts as the critical interface between the parasite and the host cell.**

This change ensures consistency in our terminology throughout the manuscript.

- Overall, there are a few concepts used throughout the text (such as autophagy, selective autophagy, canonical or non-canonical) without proper introduction. I understand this is for the sake of simplicity, but it can be lacking.

- Thank you for your valuable feedback regarding the introduction of key concepts in the manuscript. We understand the importance of ensuring that all readers, regardless of their familiarity with the field, can fully grasp the terms and concepts discussed.

In response to your concern, we have made revisions to the introduction to provide brief explanations of key concepts such as autophagy (**line 65**), selective autophagy (**line 69**), and canonical versus non-canonical signaling pathways (**line 100**). These additions are intended to give readers a clear understanding of these terms without overly expanding the text or complicating the flow of the manuscript.

We believe that these revisions will help make the manuscript more accessible to a broader audience, including those who may not be experts in parasitology or cellular signaling pathways. We appreciate your suggestion and are confident that these changes enhance the clarity and comprehensibility of our work.

- In the introduction:

o The reference number 6 appears somewhat specialized for the associated message. Could you provide justification for this choice or consider replacing it with a reference from a more general review?

Line 64

- We have removed Reference 6

o The first occurrence (not the following ones) of reference 11 is misleading because it concerns Salmonella and not Plasmodium, both of which are pathogens that generate a vacuole. If this is not a referencing issue, please clarify the context.

Line 75

- Thank you for pointing out this potential confusion. Our intention was to introduce the term "CASM" in the context of our study. To clarify and better support this introduction, we have replaced the original reference 11 with a more relevant review by Durgan et al., 2022, titled "Many roads lead to CASM." This review provides a comprehensive explanation of the CASM concept, ensuring that the context is clear and appropriately linked to the topic of our research.

We believe this change addresses the issue and provides a clearer introduction to the concept of CASM in the context of our study.

o Could you please verify the accuracy of reference 15? It appears that, even though it discusses the lysosome, it does not mention TFEB.

Line 87

- You are correct that reference 15, Kim et al., 2018, does not mention TFEB. We have removed this reference from the manuscript to ensure accuracy and relevance in our citations.

We appreciate your careful review and assistance in improving the accuracy of our manuscript.

- For the figures/results:

o It would be easier to interpret if all panels were annotated with the proteins stained (e.g., Fig. 1A: Could the authors add TFEB in red and the parasites in green, even though this information is already noted in the legend and text?). This applies to Fig. 1, 2, and S2.

Thank you for your suggestion to improve the clarity of the figures.

- We have added annotations indicating TFEB in red and *Plasmodium berghei* (*Pb*) in green to the relevant panels in Figures 1a, 2a, b, and c, specifically in the first panel of each series. This addition helps to clarify what is being shown in the images.
- We have also improved the labeling in Figure S2 to ensure consistency and ease of interpretation.

We believe these changes will make the figures more intuitive and accessible to readers. Thank you for your helpful feedback.

Figure 1 TFEB translocates to host cell nucleus upon *P. berghei* infection

Figure 2 TFEB nuclear translocation is dependent on CASM and GABARAPs

Figure S2 Starvation of Cell lines deficient for CASM

o In Fig. 1B, the panel showing the control of cells grown in full medium is missing.

- Thank you for pointing out this concern. We would like to clarify that Figure 1B already includes two panels: one labeled as "control," which shows cells grown in full medium, and another labeled "EBSS," representing cells grown under starving conditions.
- To further improve clarity, we have added a TFEB label to the figure, ensuring that it is clear which conditions each panel represents.

o In Fig. 1C, could the authors please complete the x-axis label by adding 'Pb infected' as in the graphs of Figure S1?

- We have added the label "*Pb* infected" to the x-axis in Figure 1C, as requested, to match the labeling style used in the graphs of Figure S1 (see above).

o Overall, it would be important to clarify in the **quantification** whether the authors specifically count infected cells, or they choose random fields for quantification. Additionally, the low number of infected cells in the microscopy images raises questions. It may be worthwhile to explain this situation (e.g., technical limitations, multiplicity of infection, etc.) for people who are not from the field.

- In general, we specifically search for infected cells during imaging because the infection rate in our system is quite low (1-5% of infected cells). Once we locate infected cells, we take pictures and analyze them. Importantly, we do not selectively choose certain infected cells; instead, we randomly select infected cells for analysis to avoid any potential bias. For the uninfected control, we analyze cells from the images of the same slide. Using uninfected cells from the same slides provides the best possible control, as these cells are exposed to exactly the same culture conditions as the infected cells.
- When working with transiently transfected host cells, we specifically search for cells that are transfected and infected. For the non-infected control, we analysed transfected cells on the same slide as the infected ones.

- For analyzing protein colocalization with the PVM, we use a sufficient number of representative images to calculate Pearson's correlation coefficient. Beforehand, we performed a power analysis assuming a Pearson correlation coefficient is bigger than 0.7 to estimate the number of cells needed for reliable results, and we always analyzed more cells than the power analysis suggested to ensure robust data.
- For the quantification of TFEB activity, we measured the fluorescence intensity in a defined area within the host cell nucleus and in the host cell cytoplasm and then calculated the ratio. Images were taken of infected cells, and non-infected cells on the same slide were used as controls. All cell lines used in these experiments were stably transfected with TFEBmCherry
- For quantification of mTor activity, images were taken from infected cells, and the fluorescence intensity in the cell cytoplasm was measured. Non-infected cells on the same slide served as controls.
- In case of TFEB activity in GAB3-KO cells transiently transfected with GFP-GABs, we analyzed only those cells that were both transfected and infected, using the same approach as described above. GFP positive, non-infected cells were used as controls.

o In Fig. 2B and its quantification, please consider organizing the panels in the order they are mentioned in the text for more clarity: ATG16L1 KO à ATG-16L1-DeltaWD40 àATG16L1-β.

Thank you for this suggestion to improve the clarity of our figures.

- As suggested, we reorganized the panels in Figure 2b to match the order in which they are mentioned in the text: ATG16L1 KO → ATG16L1-DeltaWD40 → ATG16L1-β We have also adjusted the order in the quantification graph in Figure 2d to maintain consistency.
- Additionally, for consistency, we have changed the order of the ATG16L1 cell lines in Figure S2b and in the quantification graph of Figure S2d to match the order in Figure 2.

Figure 2 TFEB nuclear translocation is dependent on CASM and GABARAPs

Figure S2 Starvation of Cell lines deficient for CASM

o In Fig. 3B, in my opinion, the number of replicates is a bit low, especially considering that $N > 30$ for every other experiment in the article.

Thank you for your concern regarding the number of replicates in Figure 3B. We understand the importance of ensuring sufficient replicates for robust statistical analysis.

- We performed a power analysis assuming a Pearson correlation coefficient greater than 0.7. The analysis indicated that at least 4 cells needed to be analyzed to achieve statistically significant results. In our study, we analyzed 5 cells for each condition, which meets the requirement suggested by the power analysis.

While the number of replicates in this specific experiment is lower than in others, we believe it is sufficient given the strong correlation observed and the results of the power analysis. We appreciate your attention to this detail and hope this explanation addresses your concern.

o It would be interesting for the author to justify the choice of 6 hours post-infection (hpi) for conducting the experiments.

Thank you for your question regarding the choice of 6 hours post-infection (hpi) for our experiments. This is indeed an important aspect of our study, and we appreciate the opportunity to clarify our reasoning.

- The nuclear translocation of TFEB in *Plasmodium*-infected cells is already significantly evident at 6 hpi, as shown in Figure 1C.
- While TFEB nuclear localization is most prominent at 24 hpi, this may be due to the increased parasite size and larger surface area at that later stage, which allows more FNIP1 to accumulate at the PVM. However, the underlying molecular mechanisms remain consistent, as the proteins involved are present at both 6 hpi and 24 hpi.
- In some experiments, we used transiently transfected cells. Given the low infection rate with *P. berghei* (between 1% and 5%), it was particularly challenging to obtain cells that were both transfected and infected simultaneously. We found it more feasible to achieve a reasonable number of infected and transfected cells at 6 hpi. Additionally, within the first 24 hours of infection, approximately 50% of parasites are eliminated by the host cell, further complicating the experimental setup.
- Given these challenges and for consistency, we chose to study the molecular mechanisms at 6 hpi and to conduct our microscopy quantification at this timepoint, rather than performing some experiments at 6 hpi and others at 24 hpi.
- We have added IFAs to further support our findings, demonstrating that GABARAPs localize at the parasite PVM 24 hpi. The new Figure S4c shows that endogenous GABARAPs, stained with the Ankyron, localize to the parasite's PVM at 24 hpi. Additionally, Figure S5 illustrates that ectopically expressed GFP-GABARAPs also associate with the PVM 24 hpi. These new data further confirm the role of GABARAPs in the context of *P. berghei* infection.

We hope this explanation clarifies our choice of 6 hpi as the timepoint for our experiments.

Figure S4 Endogenous GABARAPs localis to the PVM

Figure S5 Expression of GFP-GABARAPs in GAB-3KO cells